# OBJECT-AWARE INVERSION AND REASSEMBLY FOR IMAGE EDITING

**Zhen Yang**[1]  **Ganggui Ding**[1]  **Wen Wang**[1]  **Hao Chen**[1†]  **Bohan Zhuang**[2†]  **Chunhua Shen**[1]

[1] Zhejiang University, China
`{zheny.cs,dingangui,wwenxyz,haochen.cad,chunhuashen}@zju.edu.cn`
[2] Monash University, Australia
`bohan.zhuang@monash.edu`

## ABSTRACT

Diffusion-based image editing methods have achieved remarkable advances in text-driven image editing. The editing task aims to convert an input image with the original text prompt into the desired image that is well-aligned with the target text prompt. By comparing the original and target prompts, we can obtain numerous editing pairs, each comprising an object and its corresponding editing target. To allow editability while maintaining fidelity to the input image, existing editing methods typically involve a fixed number of inversion steps that project the whole input image to its noisier latent representation, followed by a denoising process guided by the target prompt. However, we find that the optimal number of inversion steps for achieving ideal editing results varies significantly among different editing pairs, owing to varying editing difficulties. Therefore, the current literature, which relies on a fixed number of inversion steps, produces sub-optimal generation quality, especially when handling multiple editing pairs in a natural image. To this end, we propose a new image editing paradigm, dubbed Object-aware Inversion and Reassembly (OIR), to enable object-level fine-grained editing. Specifically, we design a new search metric, which determines the optimal inversion steps for each editing pair, by jointly considering the editability of the target and the fidelity of the non-editing region. We use our search metric to find the optimal inversion step for each editing pair when editing an image. We then edit these editing pairs separately to avoid concept mismatch. Subsequently, we propose an additional reassembly step to seamlessly integrate the respective editing results and the non-editing region to obtain the final edited image. To systematically evaluate the effectiveness of our method, we collect two datasets called OIRBench for benchmarking single- and multi-object editing, respectively. Experiments demonstrate that our method achieves superior performance in editing object shapes, colors, materials, categories, *etc.*, especially in multi-object editing scenarios. The project page can be found here.

## 1 INTRODUCTION

Large-scale text-to-image diffusion models, such as Latent Diffusion Models (Rombach et al., 2022), SDXL (Podell et al., 2023), Imagen (Saharia et al., 2022), DALL·E 2 (Ramesh et al., 2022), have advanced significantly and garnered widespread attention. Recently, many methods have begun using diffusion models for image editing. These methods offer fine-grained control over content, yielding impressive results that enhance the field of artistic content manipulation. We focus on text-driven image editing, aiming to align the region of interest (editing region) with user-defined text prompts while protecting the non-editing region. We define the combination of the editing region and its corresponding editing target as the "editing pair". In Fig. 1, (parrot, crochet parrot) emerges as an editing pair when comparing the original prompt with target prompt 1. To enable editability in the editing region while maintaining fidelity to the input image, existing text-driven image editing methods (Tumanyan et al., 2023; Couairon et al., 2022; Hertz et al., 2022; Mokady et al., 2023;

---

[†]HC and BZ are corresponding authors.

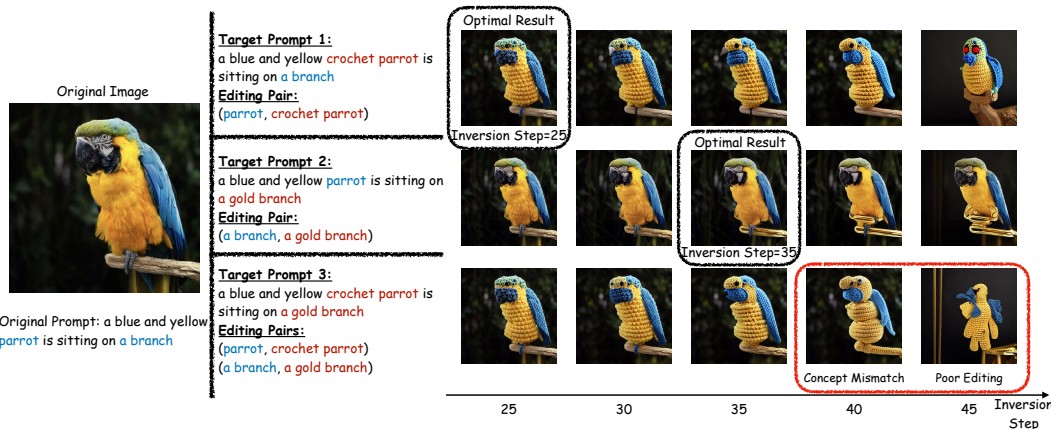

Figure 1: **Motivation.** In the process of text-driven image editing, we first inverse the original image to progressively acquire all latents. Then, we denoise each latent to generate images under the guidance of the target prompt. After obtaining all the images, the most optimally edited results are selected by human. From the first and second rows, we note that different editing pairs have unique optimal inversion steps. Moreover, we observe editing different editing pairs with the same inversion step results in *concept mismatch* or *poor editing*, as shown in the third row.

Meng et al., 2021) typically project the original image into its noisier representation, followed by a denoising process guided by the target prompt.

Our key finding is that *different editing pairs require varying inversion steps*, depending on the editing difficulties. As shown in the first and second rows in Fig. 1, if the object and target within an editing pair are similar, it requires only a few inversion steps, and vice versa. Over-applying inversion steps to easy editing pairs or insufficient steps to challenging pairs can lead to a deterioration in editing quality. This can be even worse when multiple editing pairs exist in the user prompt, as editing these objects with the same inversion step at once can lead to *concept mismatch or poor editing* in the third row of Fig. 1. However, current methods (Tumanyan et al., 2023; Couairon et al., 2022; Hertz et al., 2022; Mokady et al., 2023; Wang et al., 2023) uniformly apply a fixed inversion step to different editing pairs, ignoring the editing difficulty, which results in suboptimal editing quality.

To this end, we propose a novel method called Object-aware Inversion and Reassembly (OIR) for generating high-quality image editing results. Firstly, we design a search metric in Fig. 2. This metric automatically determines the optimal inversion step for each editing pair which jointly considers the editability of the editing object of interest and the fidelity to the original image of the non-editing region. Secondly, as shown in Fig. 3, we propose a *disassembly then reassembly* strategy to enable generic editing involving multiple editing pairs within an image. Specifically, we first search the optimal inversion step for each editing pair with our search metric and edit them separately, which effectively circumvents concept mismatch and poor editing. Afterward, we propose an additional reassembly step during denoising to seamlessly integrate the respective editing results. In this step, a simple yet effective re-inversion process is introduced to enhance the global interactions among editing regions and the non-editing region, which smooths the edges of regions and boosts the realism of the editing results.

To systematically evaluate the proposed method, we collect two new datasets containing 208 and 100 single- and multi-object text-image pairs, respectively. Both quantitative and qualitative experiments demonstrate that our method achieves competitive performance in single-object editing, and outperforms state-of-the-art (SOTA) methods by a large margin in multi-object editing scenarios.

In summary, our key contributions are as follows.

- We introduce a simple yet effective search metric to automatically determine the optimal inversion step for each editing pair, which jointly considers the editability of the editing object of interest and the fidelity to the original image of the non-editing region. The process of using a search metric to select the optimal result can be considered a new paradigm for image editing.

- We design a novel image editing paradigm, dubbed Object-aware Inversion and Reassembly, which separately inverses different editing pairs to avoid concept mismatch or poor editing and subsequently reassembles their denoised latent representations with that of the non-editing region while taking into account the interactions among them.

- We collect two new image editing datasets called OIRBench, which consist of hundreds of text-image pairs. Our method yields remarkable results, outperforming existing methods in multi-object image editing and being competitive to single-object image editing, in both quantitative and qualitative standings.

## 2  RELATED WORK

**Text-driven image generation and editing.** Early methods for text-to-image synthesis (Zhang et al., 2017; 2018b; Xu et al., 2018) are only capable of generating images in low-resolution and limited domains. Recently, with the scale-up of data volume, model capacity, and computational resources, significant progress has been made in the field of text-to-image synthesis. Representative methods like DALLE series (Ramesh et al., 2021; 2022), Imagen (Saharia et al., 2022), Stable Diffusion (Rombach et al., 2022), Parti (Yu et al., 2022), and GigaGAN (Kang et al., 2023) achieve unprecedented image generation quality and diversity in open-world scenarios. However, these methods provide limited control over the generated images. Image editing provides finer-grained control over the content of an image, by modifying the user-specified content in the desired manner while leaving other content intact. It encompasses many different tasks, including image colorization (Zhang et al., 2016), style transfer (Jing et al., 2019), image-to-image translation (Zhu et al., 2017), etc. We focus on text-driven image editing, as it provides a simple and intuitive interface for users. We refer readers to (Zhan et al., 2023) for a comprehensive survey on multimodal image synthesis.

**Text-driven image editing.** Text-driven image editing need understand the semantics in texts. The CLIP models (Radford et al., 2021), contrastively pre-trained with learning on internet-scale image-text pair data, provide a semantic-rich and aligned representation space for image and text. Therefore, several works (Abdal et al., 2020; Alaluf et al., 2021; Bau et al., 2020; Patashnik et al., 2021) attempt to combine Generative Adversarial Networks (GANs) (Goodfellow et al., 2020) with CLIP for text-driven image editing. For example, StyleCLIP (Patashnik et al., 2021) develops a text interface for StyleGAN (Karras et al., 2019) based image manipulation. However, GANs are often limited in their inversion capabilities (Xia et al., 2022), resulting in an undesired change in image.

The recent success of diffusion models in text-to-image generation has sparked a surge of interest in text-driven image editing using diffusion models (Meng et al., 2021; Hertz et al., 2022; Mokady et al., 2023; Miyake et al., 2023; Tumanyan et al., 2023; Avrahami et al., 2022; Wang et al., 2023; Brooks et al., 2023; Kawar et al., 2023). These methods typically transform an image into noise through noise addition (Meng et al., 2021) or inversion (Song et al., 2020), and then performing denoising under the guidance of the target prompt to achieve desired image editing. Early works like SDEdit (Meng et al., 2021) achieve editability by adding moderate noise to trade-off realism and faithfulness. Different from SDEdit which focuses on global editing, Blended Diffusion (Avrahami et al., 2022) and Blended Latent Diffusion (Avrahami et al., 2023) necessitates local editing by using a mask during the editing process and restricting edits solely to the masked area. Similarly, DiffEdit can automatically produce masks and considers the degree of inversion as a hyperparameter, focusing solely on the editing region. Prompt2Prompt (Hertz et al., 2022) and Plug-and-Play (PNP) (Tumanyan et al., 2023) explore attention/feature injection for better image editing performance. Compared to Prompt2Prompt, PNP can directly edit natural images. Another line of work explores better image reconstruction in inversion for improved image editing. For example, Null-text Inversion (Mokady et al., 2023) trains a null-text embedding that allows a more precise recovery of the original image from the inverted noise. Negative Prompt Inversion (Miyake et al., 2023) replaces the negative prompt with the original prompt, thus avoiding the need for training in Null-text Inversion.

While progress has been made, existing methods leverage a fixed number of inversion steps for image editing, limiting their ability to achieve optimal results. Orthogonal to existing methods, we find that superior image editing can be achieved by simply searching the optimal inversion steps for editing, without any additional training or attention/feature injection. Our approach is completely training-free and automatically searches the optimal inversion steps for various editing pairs within an image, enabling fine-grained object-aware control.

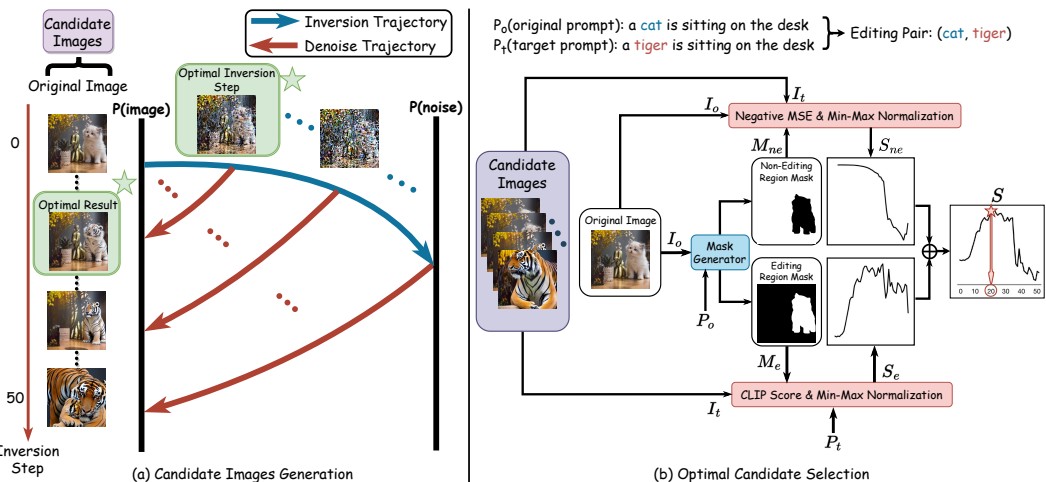

Figure 2: **Overview of the optimal inversion step search pipeline.** (a) For an editing pair, we obtain the candidate images by denoising each inverted latent. (b) We use a mask generator to jointly compute the metrics $S_e$ and $S_{ne}$, and finally we obtain $S$ by computing their average.

## 3 METHOD

In general, an image editing task can be expressed as a triplet $\langle I_o, P_o, P_t \rangle$, where $P_o$ is the original prompt describing the original image $I_o$, and $P_t$ is the target prompt reflecting the editing objective. In image editing, we aim to edit $I_o$ to the target image $I_t$ that aligns with $P_t$. To achieve this, we employ Stable Diffusion (Rombach et al., 2022), a strong text-to-image diffusion model, to enable text-driven image editing. Specifically, $I_o$ is first inverted to $I_{noise}$ using DDIM Inversion guided by $P_o$. Following that, $I_{noise}$ is denoised to generate $I_t$ guided by $P_t$ to meet the user's requirement. We further define an editing pair as $(O_o, O_t)$, where $O_o$ and $O_t$ denote an object in $P_o$ and its corresponding editing target in $P_t$, respectively. As shown in Fig. 1, there exist multiple editing pairs $\{(O_o, O_t)\}$ given an image editing task $\langle I_o, P_o, P_t \rangle$ that have multiple editing targets.

As shown in Fig. 1, each editing pair can have a distinct optimal inversion step. Hence, using a single inversion step for an image with multiple editing pairs might lead to poor editing and concept mismatch. For example, the gold branch is confusingly replaced with a crochet branch at the 40th step in the third row of Fig. 1. In Sec. 3.1, We propose a optimal inversion step search method to automatically search for the optimal inversion step for each editing pair, and in Sec. 3.2, we propose Object-aware Inversion and Reassembly (OIR) to solve the problems of poor editing and concept mismatch.

### 3.1 OPTIMAL INVERSION STEP SEARCH

**Candidate images generation.** DDIM Inversion sequentially transforms an image into its corresponding noisier latent representation. The diffusion model can construct an edited image $I_t^i$ from each intermediate result from inversion step $i$, as shown in Fig. 2 (a). This process produces a set of images $\{I_t^i\}$ called candidate images. Notably, from these candidate images, one can manually select a visually appealing result $I_t^{i^*}$ that aligns closely with $P_t$, with its non-editing region unchanged. Its associated inversion step $i^*$ is defined as the optimal inversion step. Surprisingly, comparing to the commonly used feature-injection-based image editing methods (Tumanyan et al., 2023), simply choosing a good $I_t^i$ often produces a better result. A discussion on the difference between our method and feature-injection-based image editing methods can be found in Appendix A.3.

**Optimal candidate selection.** Since manually choosing $I_t$ can be impractical, we further devise a searching algorithm as shown in Fig. 2 (b). To automate the selection process, we first apply mask generator to extract the editing region mask $M_e$ and the non-editing region mask $M_{ne}$ from $I_o$. By default, we employ Grounded-SAM (Liu et al., 2023; Kirillov et al., 2023) for mask generation. However, other alternatives can be used to obtain the editing mask, for example, we can follow

DiffEdit (Couairon et al., 2022) or MasaCtrl (Cao et al., 2023) to generate masks from the attention maps. For a detailed discussion of the mask generation process, please refer to Appendix A.5.

Subsequently, we propose a quality evaluation metric based on two criteria: $S_e$, the alignment between the editing region of the target image $I_t$ and the target prompt $P_t$; and $S_{ne}$, the degree of preservation of the non-editing region relative to the original image $I_o$.

For the first criterion regarding the editing region, we utilize CLIP score (Hessel et al., 2021) to assess alignment:

$$S_e(I_t, P_t, M_e) = \text{normalize}(\frac{\text{CLIP}_{image}(I_t,\ M_e) \cdot \text{CLIP}_{text}(P_t)}{\|\text{CLIP}_{image}(I_t,\ M_e)\|_2 \cdot \|\text{CLIP}_{text}(P_t)\|_2}), \qquad (1)$$

where $\text{CLIP}_{image}(I_t,\ M_e)$ and $\text{CLIP}_{text}(P_t)$ are the extracted editing region image feature and text feature with CLIP (Radford et al., 2021). $\text{normalize}(\cdot)$ is the min-max normalization. The normalization formula is given by: $(\{S_e\}_i - min\{S_e\})/(max\{S_e\} - min\{S_e\})$, $\{S_e\}$ denotes the complete set of $S_e$ values obtained from all candidate images, $i$ denotes the index of the image in candidate images. Insufficient inversion can restrict the editing freedom while too much inversion can lead to corrupted results. Thus, we observe that $S_e$ first rises then drops as the inversion step increases.

To measure the similarity between the non-editing regions of $I_t$ and $I_o$, we employ the negative mean squared error:

$$S_{ne}(I_t, I_o, M_{ne}) = \text{normalize}(-\| (I_t - I_o) \odot M_{ne}\|_2^2), \qquad (2)$$

where $\odot$ denotes the element-wise product, $M_{ne}$ represents the non-editing region mask. $S_{ne}$ usually decreases as inversion step grows, since the inversion process increases the reconstruction difficulty of the non-editing region. The search metric is simply an average of $S_e$ and $S_{ne}$:

$$S = 0.5 \cdot (S_e + S_{ne}), \qquad (3)$$

where $S$ is the search metric. As shown in Fig. 2 (b), we define the inversion step that has the highest search metric as the optimal inversion step.

**Acceleration for generating candidate images.** We notice that the sequential steps in generating multiple candidate images in Fig. 2 (a) are independent, and the varying number of steps make parallelization challenging. Consequently, we propose a splicing strategy over the denoising process. Firstly, we pair denoising processes of different steps to achieve equal lengths. In this way, denoising processes of the same length can proceed simultaneously for parallel acceleration, as there is no dependency between denoising processes. This strategy is detailed in Appendix A.4.

## 3.2 Object-aware Inversion and Reassembly

The optimal inversion search can be performed for each editing pair, providing us great flexibility in multi-object editing tasks. In short, to solve concept mismatch and poor editing, we disassemble the image to add different inversion noise according to the optimal step for each region and reassemble the noised regions at the corresponding stage.

**Disassembly.** From the original and target prompts, we get a sequence of editing pairs $\{(O_o, O_t)_k\}$. For preparation, we replace the entity $O_o^k$ in $P_o$ with $O_t^k$ for each pair in $\{(O_o, O_t)_k\}$, generating a sequence of guided prompts $\{P_t^k\}$, as shown in Fig. 3 (a). Then, we feed the original image and the guided prompts into the optimal inversion step search pipeline to obtain the optimal inversion step $i_k^*$ for all editing pairs, as illustrated in Fig. 3 (b). Here, each guided prompt is treated as the $P_t$ for the optimal inversion step search pipeline. Subsequently, we use the guided prompt for denoising of each editing pair, as depicted in Fig. 3 (c). Moreover, the optimal inversion step searching processes for distinct editing pairs are independent. In a multi-GPU scenario, we can run the step searching processes for different editing pairs in parallel on multiple GPUs, achieving further acceleration.

The disassembly process segregates the editing processes of different editing pairs, effectively circumventing concept mismatch. Simultaneously, this isolated denoising allows each editing pair to employ the latent from its respective optimal inversion step, thus avoiding poor editing.

**Reassembly.** In this process, given the inversion step for each editing pair, we edit and integrate the regions into the final result, as illustrated in Fig. 3 (c). We also assign an inversion step for the

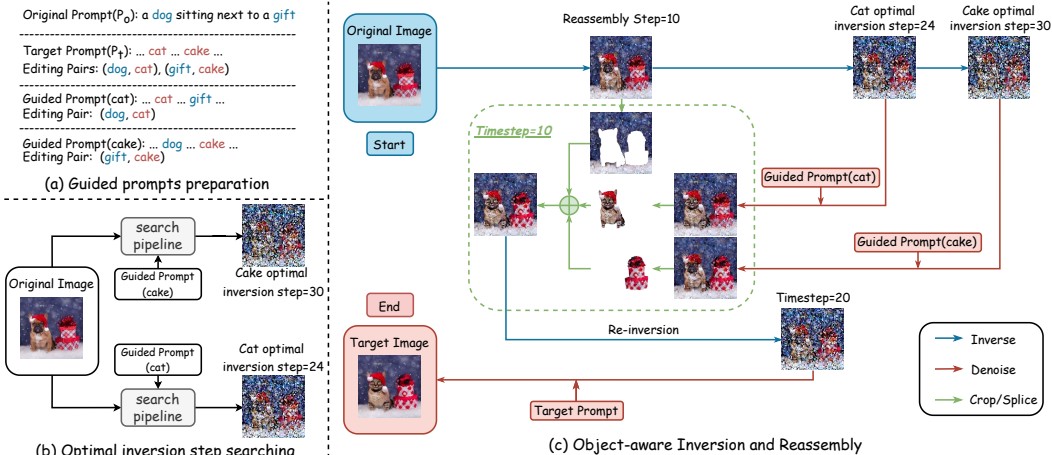

Figure 3: **Overview of object-aware inversion and reassembly.** (a) We create guided prompts for all editing pairs using $P_o$ and $P_t$. (b) For each editing pair, we utilize the optimal inversion step search pipeline to automatically find the optimal inversion step. (c) From each optimal inversion step, we guide the denoising individually using its guided prompt. We crop the denoised latent of the editing regions and splice them with the inverted latent of the non-editing region's at the reassembly step. Subsequently, we apply a re-inversion process to the reassembled latent and denoise it guided by $P_t$.

non-editing region named reassembly step $i_r$, indicating the editing regions reassemble at this step. Specifically, for the $k$-th editing region, we start from $I_{noise}^{i_k}$, the noised image at the optimal inversion step $i_k$, and use the guided prompt $P_t^k$ to denoise for $i_k - i_r$ steps. This ensures the resulting image $I_k^{i_r}$ will be at the same sampling step as the non-editing region $I_{noise}^{i_r}$. To reassemble the regions, we paste each editing result to $I_{noise}^{i_r}$ to get the reassembled image $I_r^{i_r}$ at step $i_r$. We found that for most images, setting the reassembly step to 20% of total inversion steps yields satisfactory outcomes. To enhance the fidelity of the editing results and smooth the edges of the editing region, instead of directly denoise from $I_r^{i_r}$, we introduce another noise adding process called re-inversion. Inspired by (Xu et al., 2023; Meng et al., 2023; Song et al., 2023), this process reapplies several inversion steps on the reassembled image $I_r$. In our experiments, the re-inversion step $i_{re}$ is also set to 20% of the total inversion steps, as we empirically found that it performs well for most situations. Lastly, we use the target prompt $P_t$ to guide the denoising of the re-inversion image $I_r^{i_r+i_{re}}$, facilitating global information fusion and producing the final editing result. Compared with previous methods, our reassembled latent merges the latents denoised from the optimal inversion steps of all editing pairs, along with the inverted latent from the non-edited region. This combination enables us to produce the best-edited result for each editing pair without compromising the non-editing region.

## 4 EXPERIMENTS

We evaluate our method both quantitatively and qualitatively on diverse images collected from the internet and the collection method can be found in Appendix A.1. The implementation details of our method can be found in Appendix A.2. Since single-object editing is encompassed in multi-object editing, we mainly present the experimental results on multi-object editing. Detailed results on single-object editing can be found in Appendix A.11.

### 4.1 MAIN RESULTS

**Compared methods.** We make comparisons with the state-of-the-art (SOTA) image editing methods, including DiffEdit (Couairon et al., 2022), Null-text Inversion (Mokady et al., 2023), Plug-and-Play (PNP) (Tumanyan et al., 2023), and the mask-based stable diffusion inpainting (SDI)[1].

---

[1]https://huggingface.co/runwayml/stable-diffusion-inpainting

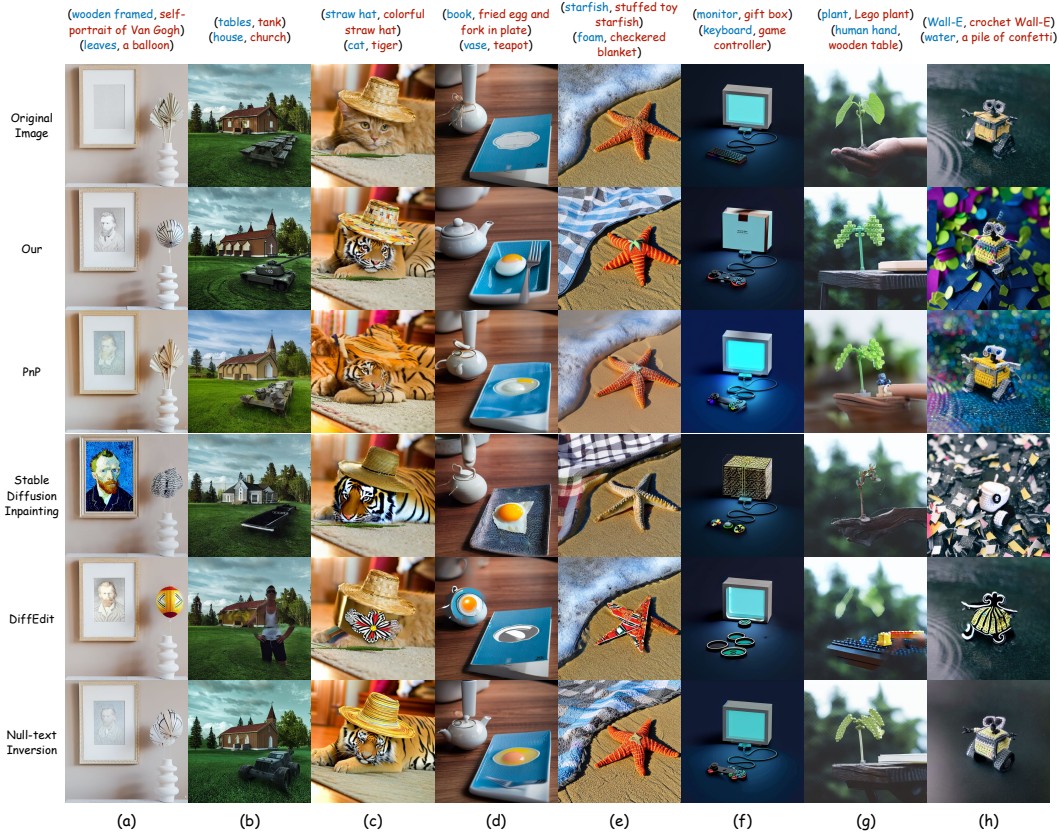

Figure 4: **Qualitative comparisons.** From top to bottom: original image, our method (OIR), PNP (Tumanyan et al., 2023), Stable Diffusion Inpainting, DiffEdit (Couairon et al., 2022), Null-text Inversion (Mokady et al., 2023). The texts at the top of the images represent editing pairs.

**Evaluation metrics.** Following the literatures (Hertz et al., 2022; Mokady et al., 2023), we use CLIP (Hessel et al., 2021; Radford et al., 2021) to calculate the alignment of edited image and target prompt. Additionally, we use MS-SSIM (Wang et al., 2003) and LPIPS (Zhang et al., 2018a) to evaluate the similarity between the edited image and the original image.

**Qualitative comparison.** We show some qualitative experimental results in Fig. 4, and additional results can be found in Appendix A.11. From our experiments, we observe the following: Firstly, SDI and DiffEdit often produce discontinuous boundaries (*e.g.*, the boundary between the tank and the grassland in Fig. 4 (b), and the desk in Fig. 4 (g)). Secondly, feature injection methods (PNP and Null-text Inversion) show better editing results in certain scenarios (*e.g.*, the Lego plant in Fig. 4 (g)). However, they overlook the variations in inversion steps for different editing pairs, leading to poor editing (*e.g.*, the foam in the Fig. 4 (e) and the monitor in Fig. 4 (f) and the water in Fig. 4 (h) are left unedited). Moreover, they face serious concept mismatch, *e.g.*, the color and texture of the "colorful strew hat" in Fig. 4 (c) is misled by the tiger's skin. Thirdly, our approach can avoid concept mismatch, since we edit each editing pair individually by disassembly (*e.g.*, the tiger and the colorful hat in Fig. 4 (c)). In addition, reassembly in our method can edit non-editing region, *e.g.*, the shadow in the background changes when the leaves turn into a balloon.

**Quantitative comparison.** We conducted quantitative analyses on our multi-object editing dataset. As illustrated in Tab. 1, we achieve state-of-the-art outcomes on CLIP score, surpassing other methods. Notably, our results show a significant improvement over the previous SOTA methods, PNP. Besides, our result on MS-SSIM is highly competitive, though it's marginally behind DiffEdit and PNP. It's worth noting that MS-SSIM primarily measures the structural similarity between the output and input images and may not always correlate with the quality of the edit. As the qualitative experiments reveal, DiffEdit and PNP occasionally neglects certain objects, leaving them unchanged, which inadvertently boosts the MS-SSIM score.

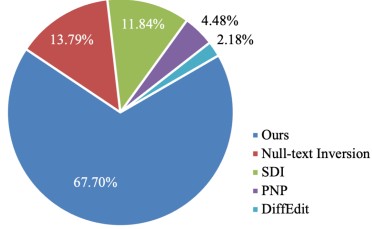

| | CLIP score ↑ | MS-SSIM ↑ | LPIPS ↓ |
|---|---|---|---|
| Ours | 30.28 | 0.653 | 0.329 |
| Plug-and-Play | 28.45 | 0.658 | 0.359 |
| SD Inpainting | 26.87 | 0.575 | 0.398 |
| DiffEdit | 20.98 | 0.736 | 0.195 |
| Null-text Inversion | 28.02 | 0.651 | 0.357 |

Figure 5: **User study results**. Users are asked to select the best results in terms of the alignment to target prompts and detail preservation of the input image.

Table 1: **Quantitative evaluation.** CLIP score measures the alignment of image and text, while MS-SSIM and LPIPS evaluate the similarity between the original and the edited images.

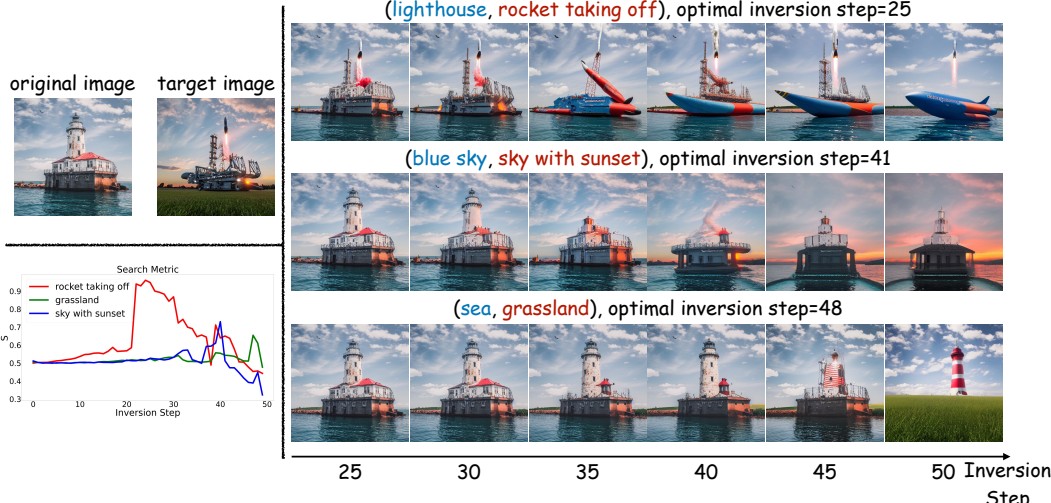

Figure 6: **Visualization of the search metric.** The images on the right represent the candidate images obtained using the search metric for each editing pair. In the bottom-left corner, curves are plotted with its x-axis representing the inversion step and the y-axis indicating the search metric $S$.

**User study.** We selected 15 images from the collected multi-object dataset for user testing, and we compared our OIR with SDI, DiffEdit, PNP, and Null-text Inversion. The study included 58 participants who were asked to consider *the alignment to the target prompt* and *preservation of details of the original image*, and then select the most suitable image from a set of five randomly arranged images on each occasion. As can be seen from Fig. 5, the image generated by OIR is the favorite of 66.7% of the participants, demonstrating the superiority of our method. An example of the questionnaire can be found in Appendix A.11.

## 4.2 VISUALIZATION OF THE SEARCH METRIC

We visualize the candidate images and their search metric in Fig. 6 to gain a clearer understanding of the trend of the search metric $S$. It's evident that each editing pair has its own optimal inversion step, with significant variations between them. For instance, the (sea, and grassland) perform optimally between steps 45 and 50. Meanwhile, the (lighthouse, rocket taking off) is most effective around the 25th step, but experience significant background degradation after the 35th step. As shown in the curves in Fig. 6, the optimal inversion step selected by our search metric aligns closely with the optimal editing results, showcasing the efficacy of our approach. In addition, in the curves of each editing pair, we observe a trend that the search metric first increases and then decreases as the inversion step increases. The reasons are as follows: When the inversion step is small, the background changes slightly, making editing region alignment with the target prompt the dominant

factor in the search metric. As the inversion step grows, the edited result aligns well with the target prompt, amplifying the influence of background consistency in the search metric. More visualization results can be found in Appendix A.11.

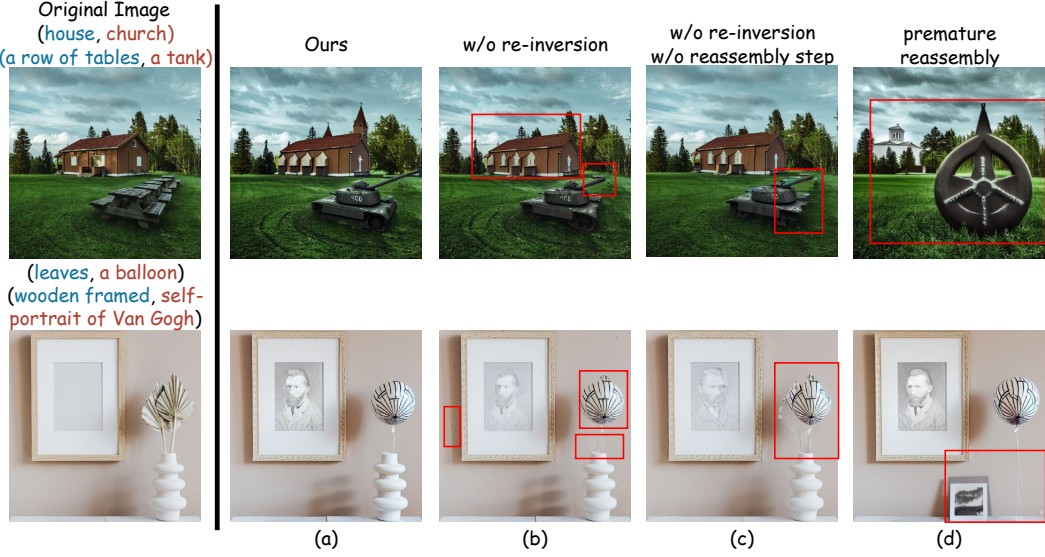

Figure 7: **Ablations for OIR.** The images and texts on the far left are the original images and their editing pairs. The remaining images represent the results after ablating the OIR. The editing effect within the red box is poor.

### 4.3 ABLATION STUDY

As shown in Fig. 7, we conduct ablation experiments on each module in the OIR. Initially, we set the reassembly step to a significantly large value, essentially merging editing pairs at an early stage. As observed in Fig. 7(d), mismatch emerge between different concepts, such as the area designated for the house being overtaken by the trees in the background. Additionally, as depicted in Fig. 7(b), the image edges become notably rough, unrealistic, and contain noise, when re-inversion is omitted. Without re-inversion, different regions are denoised independently, leading to a weaker representation of the relationships between them. If neither is added, not only the concept mismatch, but the edges are sharp and noisy, as shown in Fig. 7(c).

## 5 CONCLUSION AND FUTURE WORK

We have proposed a new search metric to seek the optimal inversion step for each editing pair, and this search method represents a new paradigm in image editing. Using search metric, we present an innovative paradigm, dubbed Object-aware Inversion and Reassembly (OIR), for mulit-object image editing. OIR can disentangle the denoising process for each editing pair to prevent concept mismatch or poor editing and reassemble them with the non-editing region while taking into account their interactions. Our OIR can not only deliver remarkable editing results within the editing region but also preserve the non-editing region. It achieves impressive performance in both qualitative and quantitative experiments. However, our method requires additional inference time for optimal inversion step search, and the effectiveness of our approach on other generic editing tasks, such as video editing, remains to be verified. Furthermore, exploring the integration of OIR with other inversion-based editing methods is also an area worth investigating. We consider addressing these issues in future work.

## 6 ACKNOWLEDGMENTS

This work was supported by National Key R&D Program of China (No. 2022ZD0118700). The authors would like to thanks Hangzhou City University for accessing its GPU cluster.

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

# A  APPENDIX

## A.1  DATA COLLECTION

To assess the effectiveness of real-world image editing, we collect two datasets by carefully selecting images from various reputable websites, namely Pexels [2], Unsplash [3], and 500px [4]. We use the first dataset to test the ability for single-object editing of the search metric, including animals, vehicles, food, and more. The second dataset is created to evaluate the method's multi-object editing capabilities. Each photo in this dataset contains two editable objects. We also designed one or more prompts for each image to test the editing effectiveness. The images resize to 512x512 pixels.

## A.2  IMPLEMENTATION DETAILS

We use Diffusers[5] implementation of Stable Diffusion v1.4 [6] in our experiments. For DDIM Inversion, we used a uniform setting of 50 steps. Our method employs the simplest editing paradigm, consisting of first applying DDIM Inversion to transform the original image into a noisy latent, and then conducting denoising guided by the target prompt to achieve the desired editing effect. We employ the CLIP base model to compute the CLIP score as outlined by (Hessel et al., 2021) for our search metric, and utilize the CLIP large model for quantitative evaluation. Following (Miyake et al., 2023), we set the negative prompt as the original prompt for the denoising process throughout our experiments. We use Grounded-SAM[7] to generate masks. We use our search metric to perform single-object editing and compare with Plug-and-Play (PNP) (Tumanyan et al., 2023), Stable Diffusion Inpainting (SDI)[8], and DiffEdit (Couairon et al., 2022). For multi-object editing, we compare our method to PNP, SDI, DiffEdit, and Null-text Inversion (Mokady et al., 2023), which can directly support multi-object editing.

In the single-object editing experiments, the parameters of PNP[9] were kept consistent with the default values specified in the code. DiffEdit[10] utilized the default parameters from the diffusers library. SDI utilized the code from the Diffusers. The random seed is set to 1 for all experiments.

In the multi-object editing experiments, PNP can easily be extended to generalized multi-object editing scenarios. For SDI, we consider three approaches to extend it to multi-object scenarios. *Method 1* uses a mask to frame out all editing regions and use a target prompt to guide the image editing. In *Method 2*, different masks are produced for different editing regions. These regions then utilize guided prompts for directed generation. Subsequently, after cropping, the results are seamlessly merged together. In the third approach, we substitute the guided prompt from *Method 2* with $O_t$ specific to each editing pair. *Method 3* is used in Fig. 4 because it has the best visual effects. All our experiments are conducted on the GeForce RTX 3090.

## A.3  SCHEMATIC COMPARISON

The automatic image selection through search metric is a new image editing paradigm, which is theoretically similar to the feature-injected-based image editing method. We use the most representative PNP among feature-injected-based image editing methods as an example. In the scenario of 50 steps of DDIM Inversion, PNP will select the latent after 50 steps of inversion, as shown in Fig. 8 (a). At this time, latent is the most noisey and has the greatest editability. If we directly denoise the latent, it will severely destroy the layout of the original image. To solve this problem, PNP reduces the editing freedom by injecting features. Compared with PNP, our search metric in Fig. 8 (b) automatically selects the most suitable latent by controlling the number of inversion steps to achieve editing fidality and editability.

---

[2]https://www.pexels.com/zh-cn/

[3]https://unsplash.com/

[4]https://500px.com/

[5]https://github.com/huggingface/diffusers

[6]https://huggingface.co/CompVis/stable-diffusion-v1-4

[7]https://github.com/IDEA-Research/Grounded-Segment-Anything

[8]https://huggingface.co/runwayml/stable-diffusion-inpainting

[9]https://github.com/MichalGeyer/pnp-diffusers

[10]https://huggingface.co/docs/diffusers/api/pipelines/diffedit

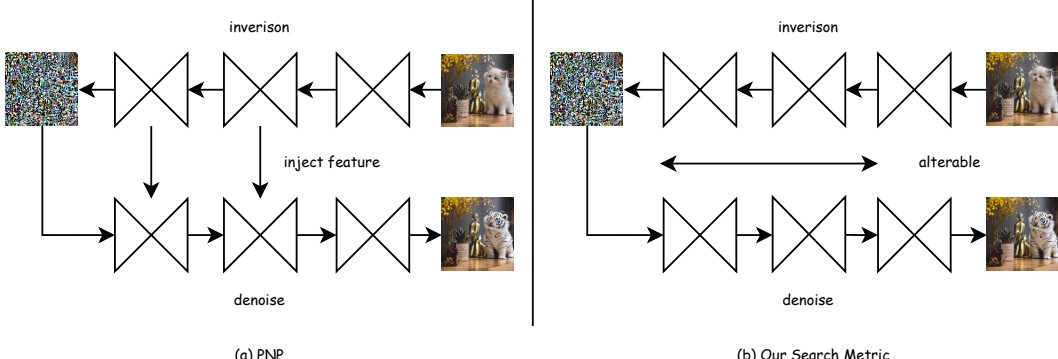

Figure 8: Left: the process of the feature injection method. Right: the process of our search metric.

## A.4 ACCELERATION FOR GENERATING CANDIDATE IMAGES

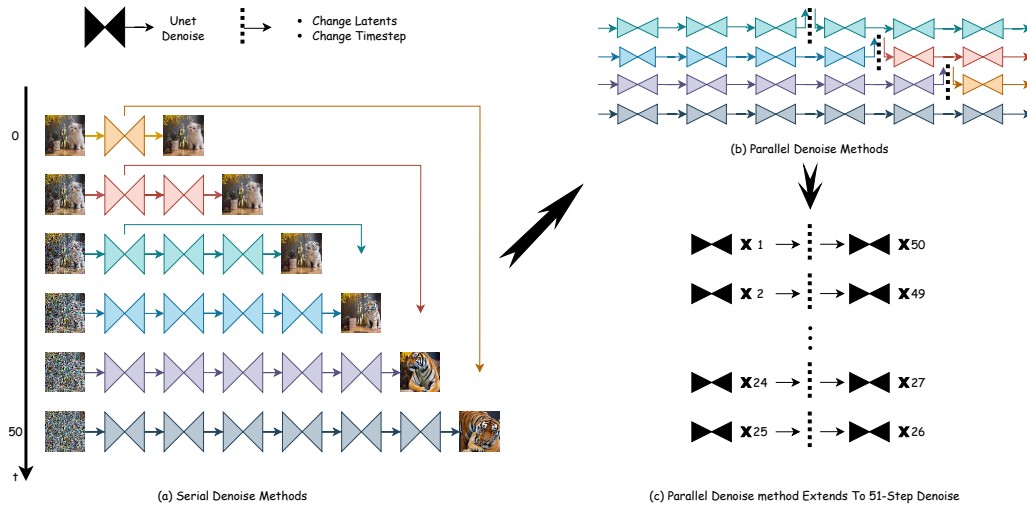

Figure 9: The funnel shape represents the denoising process, while the vertical bold lines represent the operations of changing the latent and changing the timestep. (a) Schematic for generating all target images. (b) Our proposed method for implementing parallel generation of all target images. (c) Extending the methodology to the 50-step DDIM Inversion.

As shown in Fig. 9 (a), generating candidate images is a serial process and there is no interdependence in different denoise processes. We leverage this characteristic to propose an acceleration method for generating candidate images, illustrated in Fig. 9 (b). This method involves equalizing the length of denoise operations and introducing "change latent" and "change timestep" operations at the junctions. By denoising all latents simultaneously, we will change the generation speed of candidate images to the same speed as generating a picture. An extension of our approach, tailored to the context where DDIM Inversion spans 50 steps, is shown in Fig. 9 (c).

## A.5 GENERATING MASK AND VISUAL CLIP FEATURE

We utilize the Grounded-SAM (Liu et al., 2023; Kirillov et al., 2023) to generate masks of the editing regions and we will use these masks to compute the CLIP score (Hessel et al., 2021). The detailed process is depicted in Fig. 10 (a), and examples of the segmentation result is presented in Fig. 11. Since only the object features within the mask region are of interest, a self-attention mask is applied to restrict the feature extraction of CLIP vision model. The mask is resized to match the number of patches in CLIP and is then transformed into an attention mask as depicted in Fig. 10 (b). Finally, it is fed into the self-attention of the CLIP vision model for interaction with the original image.

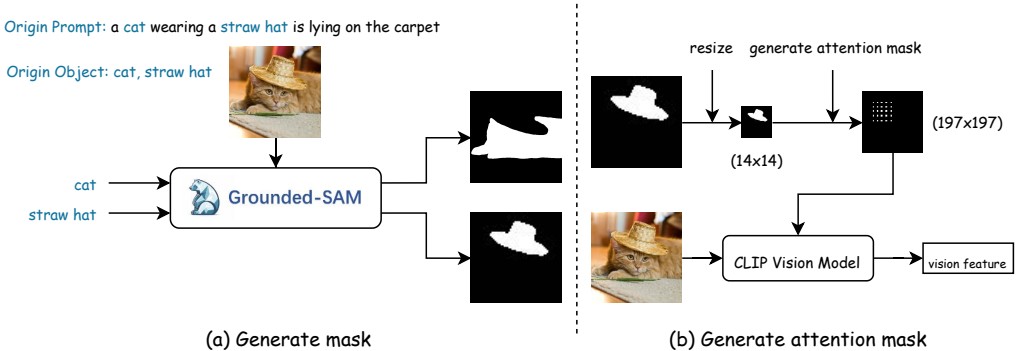

Figure 10: Left: the process of generating editing region mask. Right: the process of generating CLIP's self-attention mask through object mask.

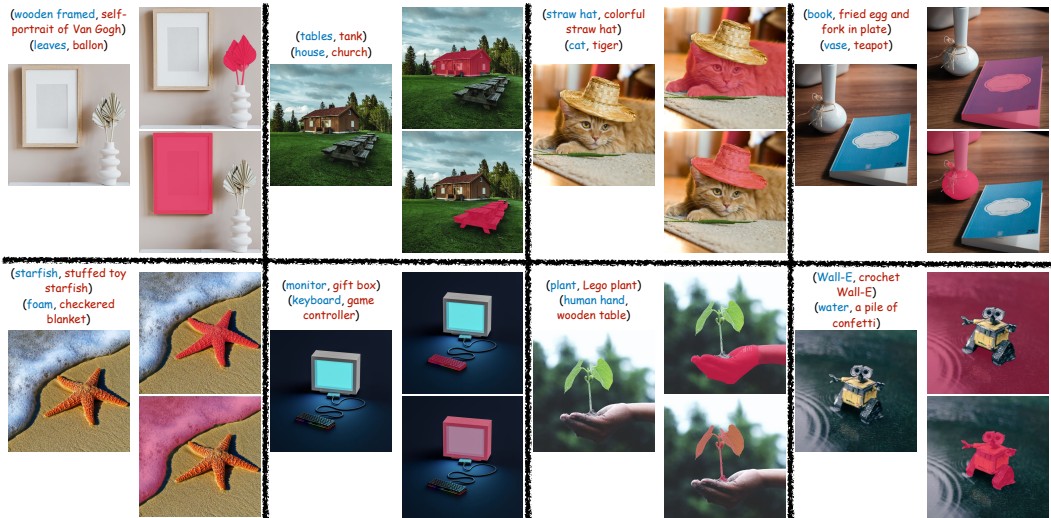

Figure 11: Segmentation masks for images in Fig. 4.

## A.6 COMPARISON OF EDITING SPEED

| Method | Ours OIR (Multi-GPU) | Ours OIR (Single-GPU) | Null-text Inversion | Plug-and-Play | Stable Diffusion Inpainting | DiffEdit |
|---|---|---|---|---|---|---|
| **Average time cost** | 155s | 298s | 173s | 224s | 6s | 15s |
| **Maximum GPU usage** | 19.75 GB | 19.75 GB | 22.06 GB | 7.31 GB | 10.76 GB | 13.51 GB |

Table 2: Editing speed and maximum GPU usage of different editing methods in multi-object editing.

We evaluate the speed of diverse editing techniques applied to a multi-object dataset using the GeForce RTX 3090, with the results detailed in Table 2. We ignore the time overhead for pre- and post-processing, such as model loading and file reading, concentrating primarily on computational costs. "Time cost" denotes the expended time on editing an image, and "Maximum GPU usage" represents the peak GPU utilization by a single GPU during the editing process. Our OIR implementation uses the acceleration scheme in Appendix A.4. In Tab. 2, OIR (Multi-GPU) indicates running OIR on two GPUs, while OIR (Single-GPU) runs the same process on a single GPU, searching the optimal inversion step for different editing pairs sequentially. We use the default hyperparameters for Null-text Inversion (NTI), Plug-and-Play (PNP), and DiffEdit, using their open-source codes. We can observe that, although OIR is slower than NTI and PNP on a single GPU, our method excels in editing capability compared to these methods. Additionally, the additional time overhead is

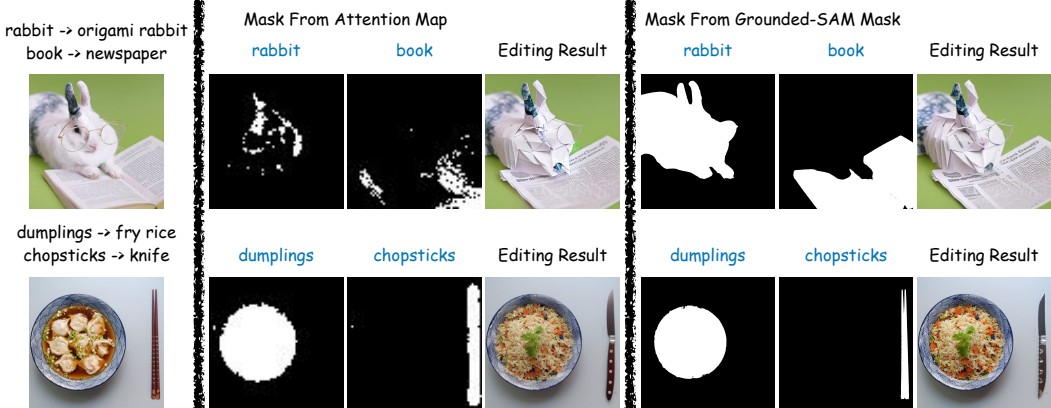

Figure 12: Compare the impact of different mask generators in the search metric on editing results.

within an acceptable range. Moreover, our method can be accelerated significantly when running on multi-GPUs, outperforming NTI and PNP in speed, where NTI and PNP do not have clear solutions that can be accelerated on multi-GPU due to the temporal dependency between the denoise steps.

### A.7 Comparison of different mask generators

We compare the influence of different mask generators on OIR, as shown in Fig. 12. During our testing, we employ two types of mask generators. The first approach is the Grounded-SAM method within the segment model. The second approach involves extracting masks using the attention map from Stable Diffusion, following methods like DiffEdit (Couairon et al., 2022) and MasaCtrl (Cao et al., 2023). Specifically, We employ the mask extraction method from DiffEdit, which eliminates the need for introducing an additional model. The first line in Fig. 12 reveals a notable accuracy loss in the mask extracted from the attention map compared to the one extracted by Grounded-SAM. Nevertheless, OIR consistently produces excellent editing results with these sub-optimal masks, indicating the robustness of our method across various mask generators. Moreover, as seen from the second line in Fig. 12, our method performs well when using the mask extracted from the attention map. Thus, our approach is not reliant on the segment model, highlighting its robustness in handling different masks and producing plausible editing results.

### A.8 The combination of Search Metric and other inversion-based image editing methods

Our search metric can be used in conjunction with other inversion-based image editing methods. Here we use the fusion of Null-text Inversion (NTI) Mokady et al. (2023) and the search metric as an example. In NTI, the "cross replace step" is a crucial hyperparameter that determines the proportion of feature injection. A higher value for the "cross replace step" retains more of the original image information, while a lower value allows for more freedom in editing. In the NTI's open-source code, the "cross replace step" is set to 0.8. There are multiple ways to combine the search metric and NTI. The first approach is to fix the inversion step and use the search metric to find the optimal "cross replace step". The second approach is to fix the "cross replace step" and use the search metric to find the optimal inversion step. The third approach involves simultaneously searching for both the "cross replace step" and inversion steps. Fig. 13 shows the experimental results for the first alternative. From the results in the first row, it is clear that the "cross replace step" in the official code fails to transform the wooden house into a glass house. By contrast, by exploring the parameters of the search metric, we can achieve improved editing results. As can be seen from the second row in Fig. 13, it is evident that the "cross replace step" varies significantly for different editing tasks, making manual adjustment impractical. Therefore, the search metric is highly valuable in this context. Additionally, the search metric can be used as an ensemble learning approach. For example, if three editing methods are applied simultaneously, each producing different editing results, the search metric can be used to select the optimal result as the final editing outcome.

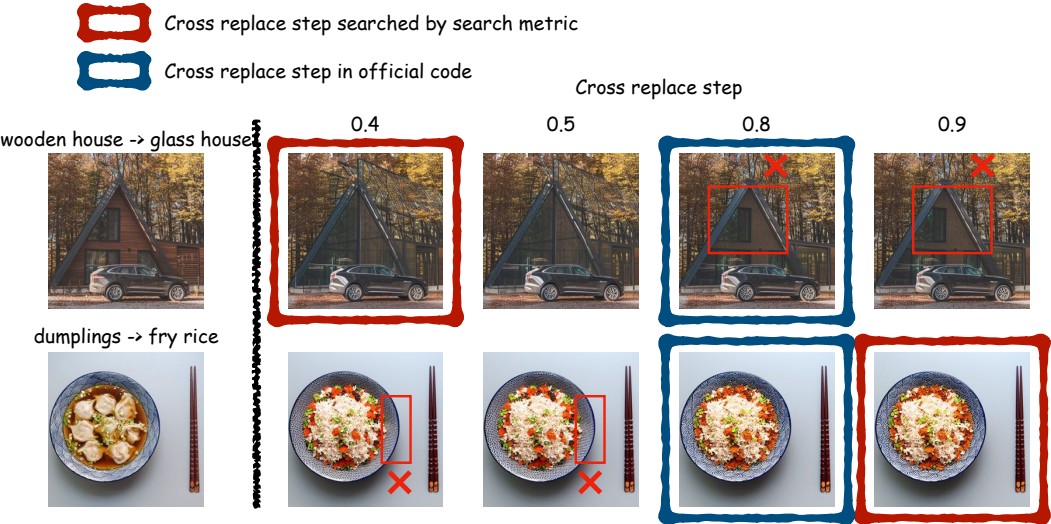

Figure 13: The combination of Search Metric and Null-text Inverison.

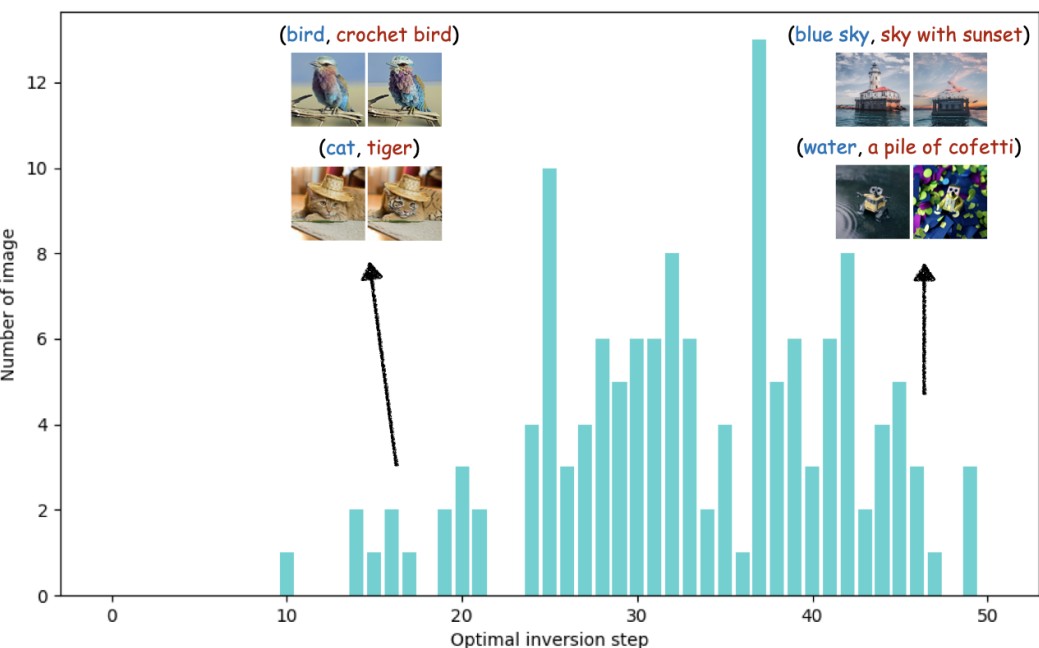

Figure 14: The distribution of optimal inversion steps in multi-object dataset.

## A.9  THE DISTRIBUTION OF OPTIMAL INVERSION STEPS IN MULTI-OBJECT DATASET

To determine the distribution of optimal inversion steps in images, we use the search metric to find the optimal inversion steps for 200 editing pairs in 100 images. The results of these editing pairs are shown in Fig. 14. This figure illustrates the number of optimal inversion steps on the horizontal axis for multi-object images, while the vertical axis represents the number of images corresponding to each optimal inversion step. From Fig. 14, it is clear that different editing targets require different optimal inversion steps. We notice that larger optimal inversion steps are necessary when altering backgrounds or objects with significant shape changes, such as the sky or the ground. Conversely, scenarios with smaller inversion steps typically involve objects and targets with similar shapes.

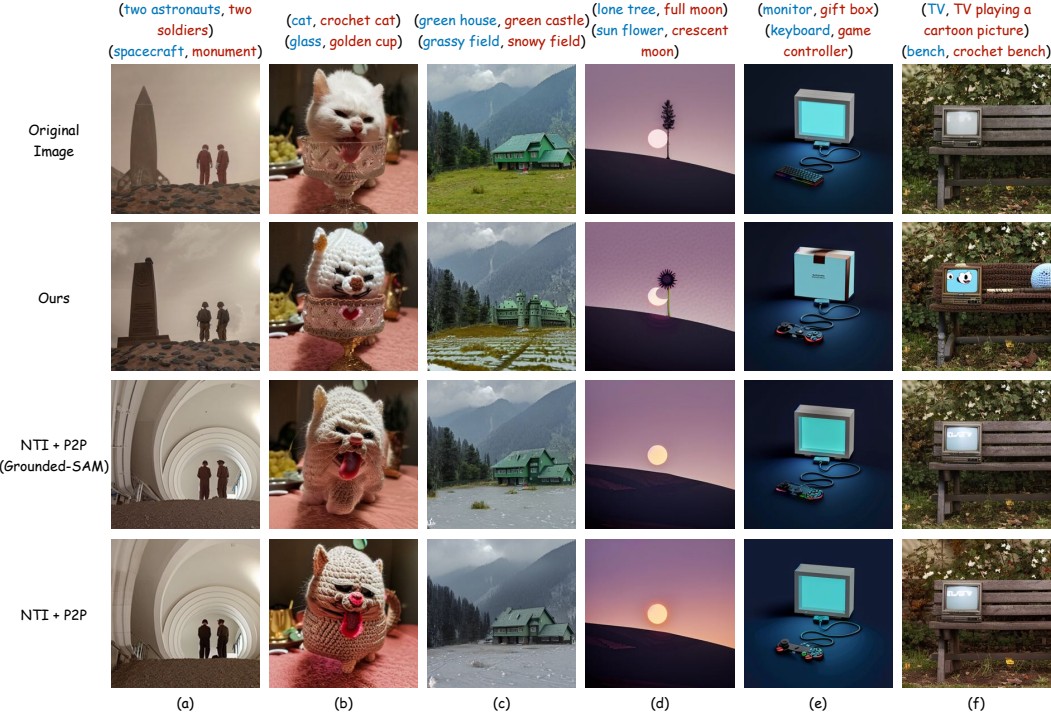

Figure 15: Our OIR vs. Null-text Inversion with Grounded-SAM.

## A.10    OIR vs. Null-text Inversion with Grounded-SAM

Null-text Inversion (NTI) (Mokady et al., 2023) is combined with Prompt-to-Prompt (P2P) (Hertz et al., 2022) by default, which utilizes the attention map to extract masks and improve background preservation, allowing for local edits. We replace the mask generation method with Grounded-SAM to examine whether a precise mask extractor would enhance the editing effectiveness of NTI. In Fig. 15, columns a, b, and c use the word swap with local edit approach from P2P. Due to the different lengths of the original prompt and target prompt, columns d, e, and f in Fig. 15 utilize the prompt refinement with local edit method from P2P. From columns a and d in Fig. 15, we notice that NTI with Grounded-SAM fails to preserve the layout information of the original image. From column b, it is evident that NTI cannot effectively address the concept mismatch. From columns c, e, and f in Fig. 15, it can be seen that NTI fails to overcome the issue of poor editing. The main reason for the poor performance is that NTI does not take into account that different editing pairs for the same image should have distinct optimal inversion steps. What's more, OIR is training-free, while NTI requires additional training.

## A.11    Additional results

| | CLIP score ↑ | MS-SSIM ↑ | LPIPS ↓ |
|---|---|---|---|
| Search Metric | 27.49 | 0.719 | 0.215 |
| Plug-and-Play (Tumanyan et al., 2023) | 27.39 | 0.680 | 0.293 |
| Stable Diffusion Inpainting | 27.36 | 0.706 | 0.180 |
| DiffEdit (Couairon et al., 2022) | 21.23 | 0.726 | 0.200 |

Table 3: **Quantitative evaluation for search metric on single-object editing.** We use CLIP (Hessel et al., 2021; Radford et al., 2021) to calculate the alignment of image and text, and use MS-SSIM (Wang et al., 2003) and LPIPS (Zhang et al., 2018a) to evaluate the similarity between the target image and the original image.

We compared the single-object editing capabilities of our search metric with the state-of-the-art (SOTA) method, as shown in Fig. 16. We have provided quantitative metrics for our single-object

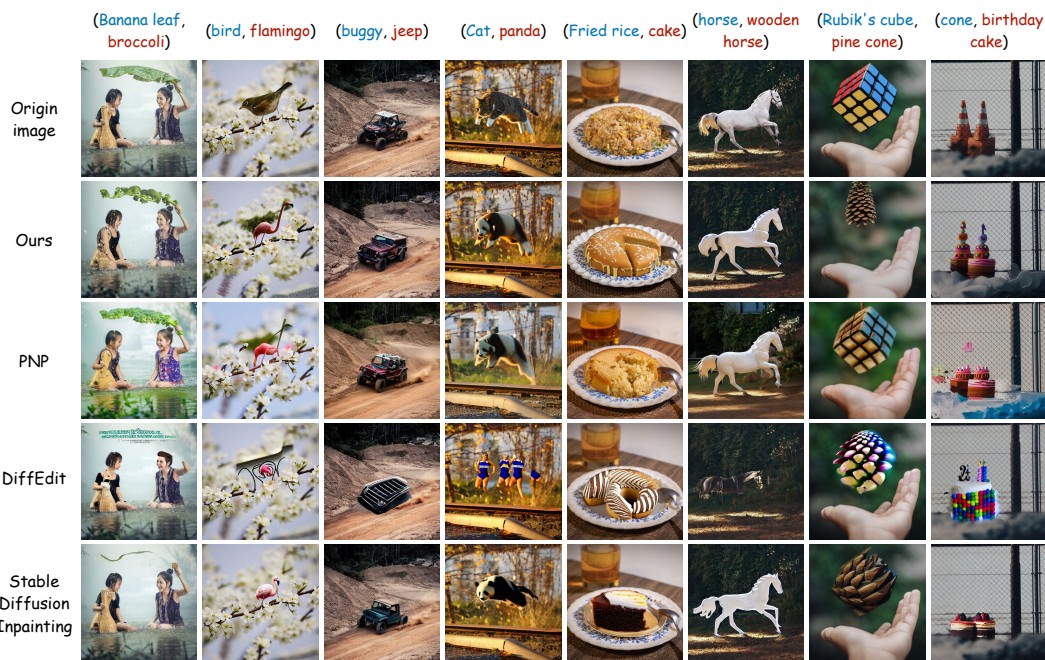

Figure 16: Qualitative comparison on the search metric.

dataset in Tab. 3, and it's evident that our method is comparable to the current SOTA approach. Simultaneously, we display numerous OIR results on the multi-object dataset, as depicted in Fig. 17 and Fig. 18. The comparison between OIR and SDI's three methods is shown in Tab. 4. Additionally, we have included some search metric visualization experiments, as presented in Fig. 19. The visualization of the optimal results for different editing pairs in Fig. 4 can be seen in Fig. 20. Fig. 21 displays our user study questionnaire form.

|  | CLIP score ↑ | MS-SSIM ↑ | LPIPS ↓ |
|---|---|---|---|
| OIR | 34.98 | 0.653 | 0.329 |
| Stable Diffusion Inpainting (Method1) | 32.97 | 0.516 | 0.420 |
| Stable Diffusion Inpainting (Method2) | 32.02 | 0.581 | 0.386 |
| Stable Diffusion Inpainting (Method3) | 32.16 | 0.575 | 0.398 |

Table 4: **Quantitative evaluation for OIR with Stable Diffusion Inpainting on multi-object editing.** We use CLIP (Hessel et al., 2021; Radford et al., 2021) to calculate the alignment of image and text, and use MS-SSIM (Wang et al., 2003) and LPIPS (Zhang et al., 2018a) to evaluate the similarity between the target image and the original image.

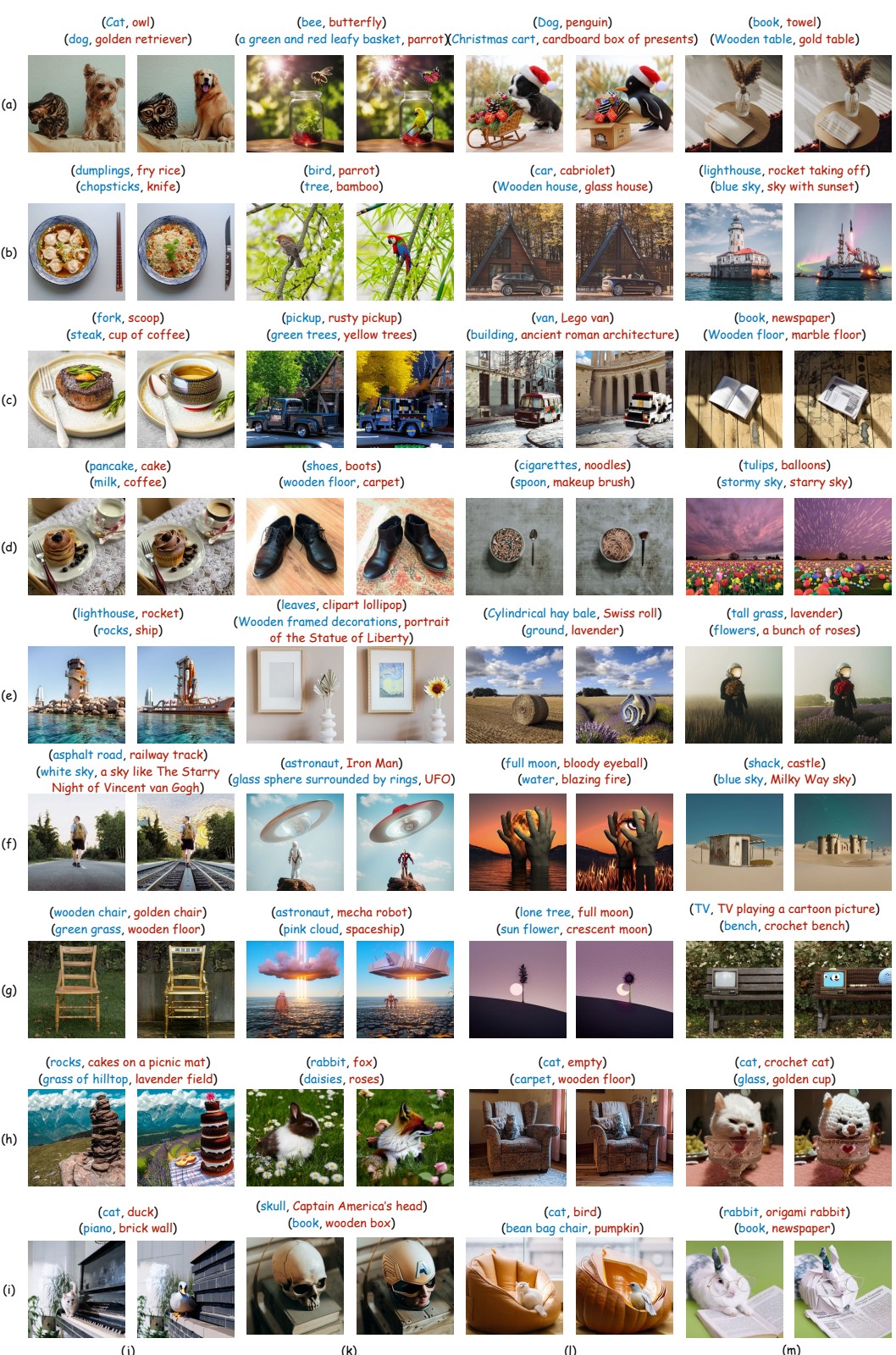

Figure 17: Additional qualitative results for OIR. It's evident that our method can edit not only objects but also backgrounds, including the sky and ground, and facilitate style transfer. Examples like (b, k), (b, m), (c, l), (c, m), (d, m) involve background editing, (c, k) encompasses seasonal editing, and (f, j) achieves style transfer.

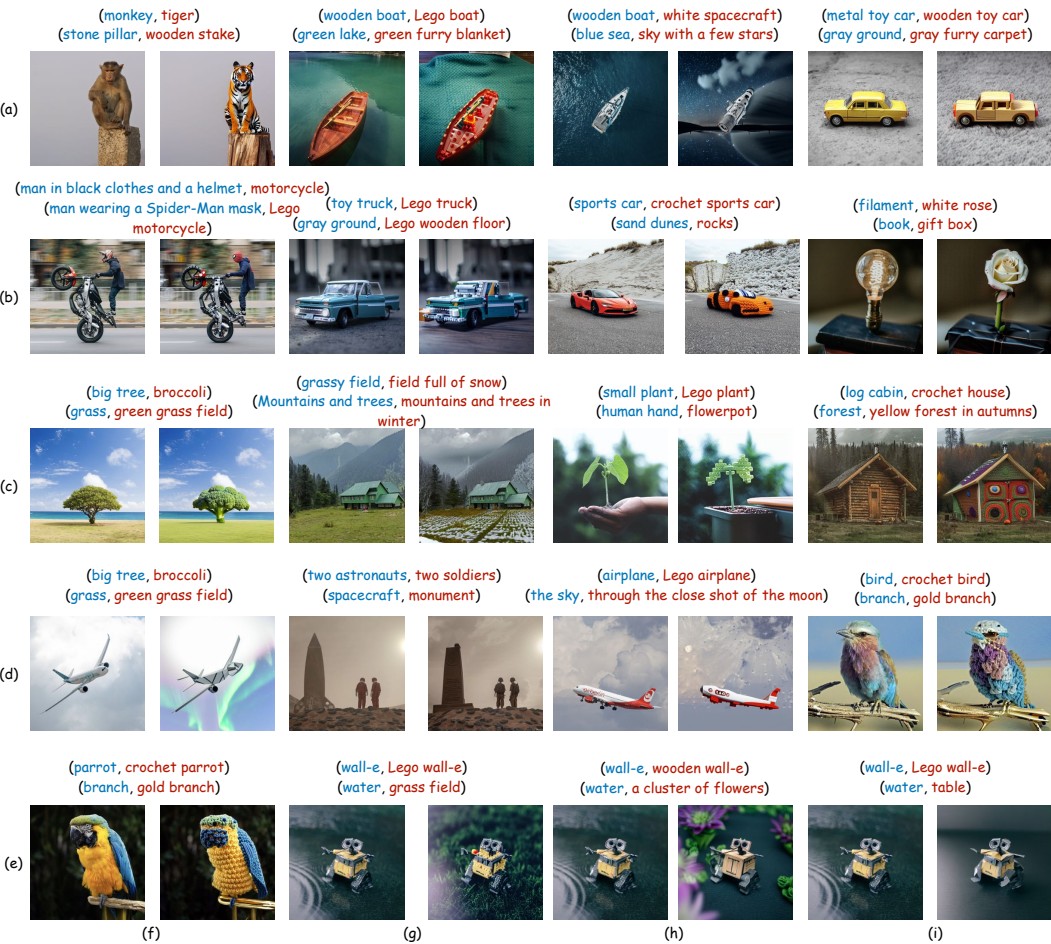

Figure 18: Additional qualitative results for OIR. It's evident that our method can edit not only objects but also backgrounds, including the sky and ground, and facilitate style transfer. Examples like (a, h), (c, f), (d, f), (d, h), (e, g), (e, h), (e, i) involve background editing. (c, g), (c, i) encompasses seasonal editing.

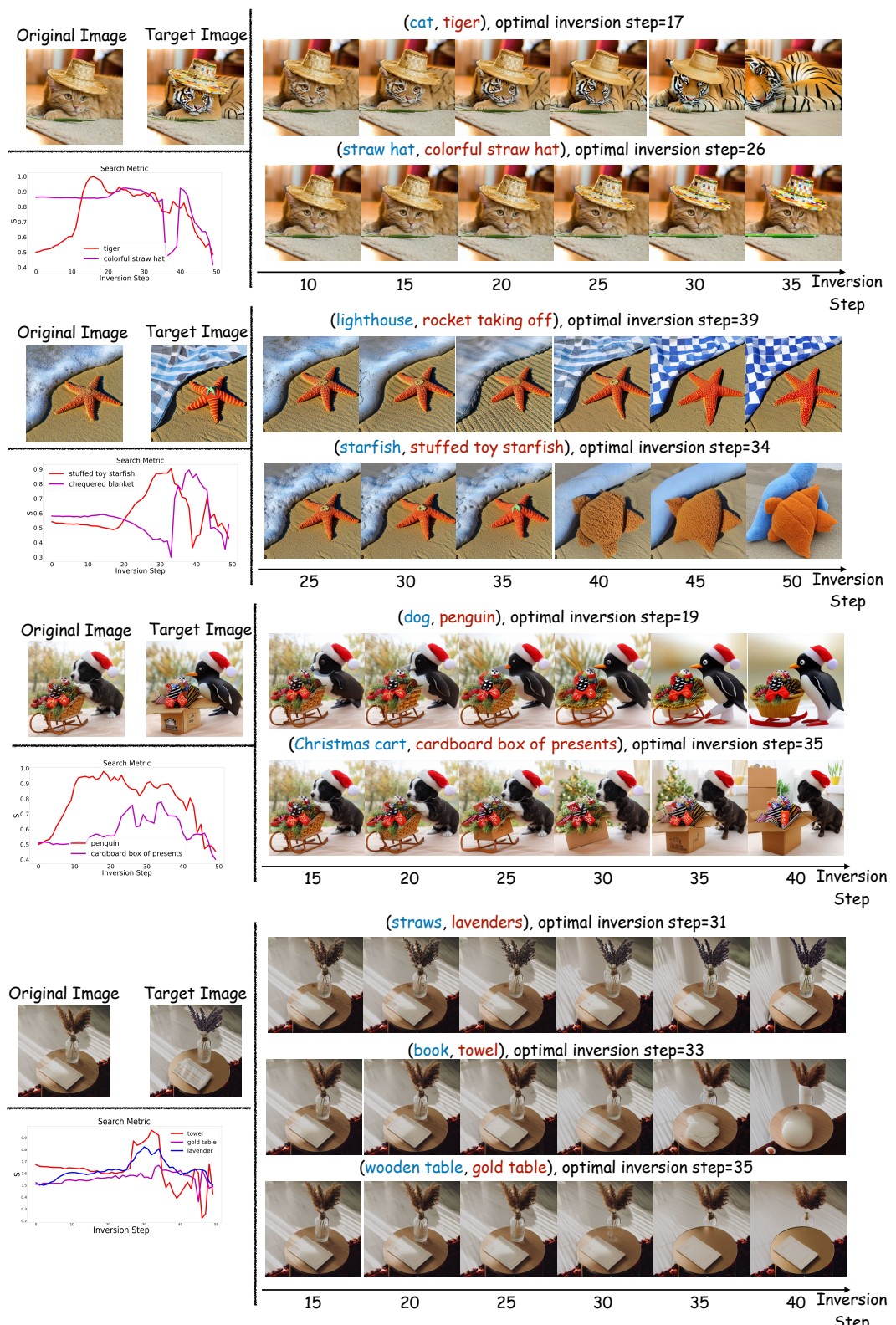

Figure 19: Additional visualization results of our search metric.

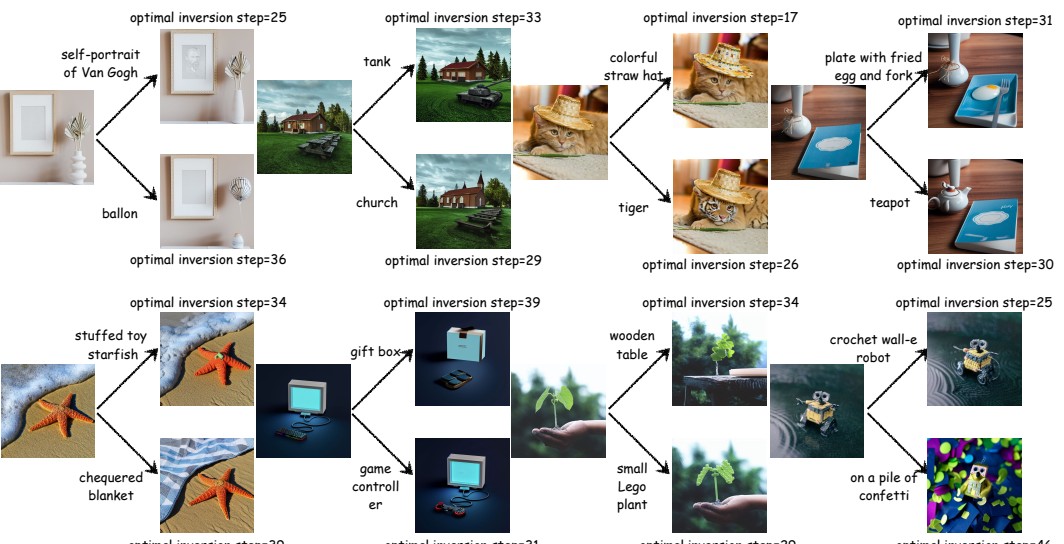

Figure 20: The optimal editing results for each editing pair in Fig. 4.

*16. The following shows the original image the editing results of 5 image editing methods.

Please select the result that best aligns the target prompt and preserves details of the original image.

Target Prompt:
1. turn 'vase' into 'teapot'
2. turn 'book' into 'plate with fried egg and fork'

original image：

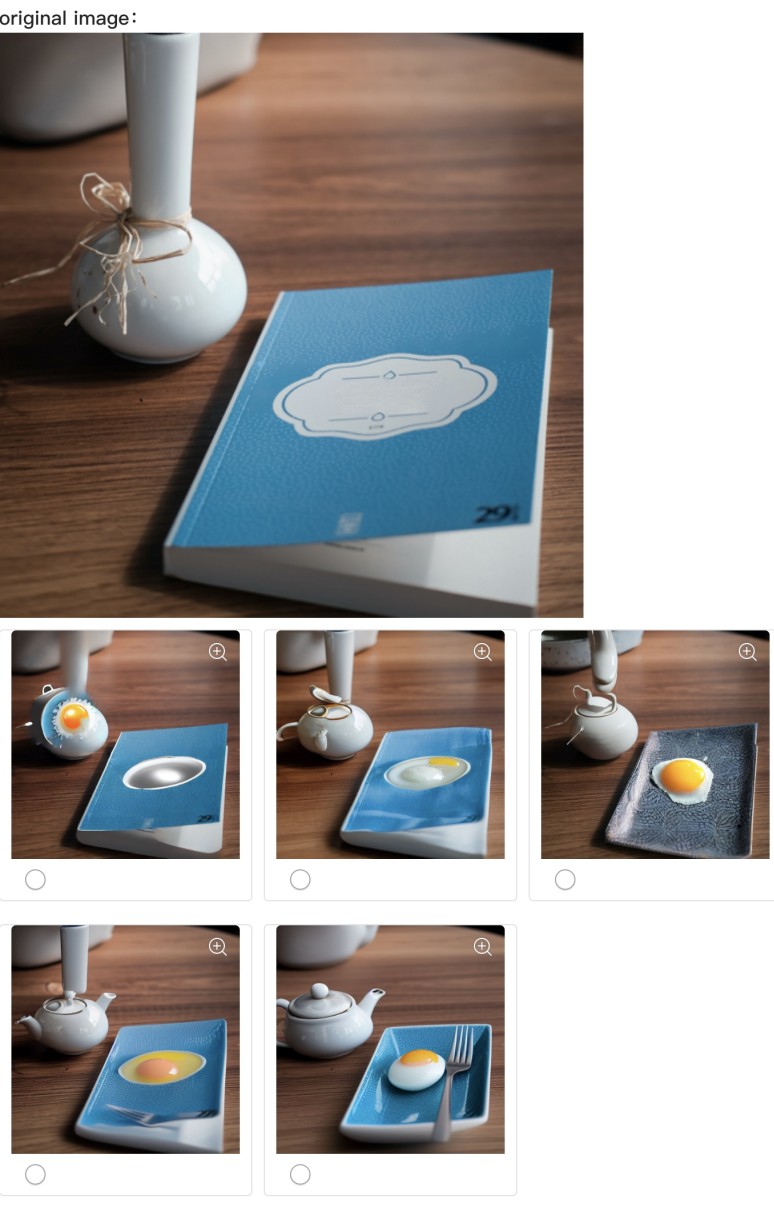

Figure 21: User study print screen.

