# OpenReview forum: "Object-Aware Inversion and Reassembly for Image Editing"
_ICLR.cc/2024/Conference — ICLR 2024 poster_

### Official Review · Reviewer_nVsr · 2023-10-30

**Soundness:** 4 excellent
**Presentation:** 3 good
**Contribution:** 3 good
**Rating:** 8
**Confidence:** 5

**Summary:**

This paper achieves high-quality object-level image editing through a simple yet effective method. Authors find that the editing for different parts requires different noising levels used to inverse the given image to latent space. As such, they search for optimal inversion steps for different parts based on a search metric for both the edited and non-edited regions respectively. In order to produce a natural final result, the results of different parts are seamlessly blended, which yields a harmonic and globally coherent editing output. Albeit its simplicity, the proposed method outperforms competitive works when editing real images and shows promise in real usage.

**Strengths:**

- This paper identifies an interesting and useful phenomenon that the structure of different objects corresponds to a different level of latent as modeled in the diffusion process. To achieve the optimal trade-off between semantic preservation and text-based editing, the optimal number of inversion steps can be determined based on a quantitative measure.

- The image editing results are indeed impressive, as shown in the main text and the appendix. Both the quantitative and qualitative results demonstrate the advantage over strong prior works.

- Since the denoising process of different image parts is disentangled, the proposed method supports fine-grained multiple object editing, which is not featured in prior works.

- The parallel denoising strategy effectively reduces the search speed.

**Weaknesses:**

- One major drawback of the method is the search speed. Since the inversion steps of each editing part should be comprehensively searched, this may pose a challenge when editing for multiple regions. It would be interesting to conduct a coarse-to-fine search strategy for speedup. Moreover, in the paper, it is suggested to report the image editing speed for different methods since some other baselines, like prompt-to-prompt, can edit images in a feed-forward manner.

- Some writing parts can be improved for better clarity. For example, what's the detailed formulation of min-max normalization? It is suggested to better rephrase the term "concept mismatch".

- Since the re-inversion step is always set to 20% of the total steps, the non-edit regions will inevitably be affected. Such subtle change is particularly apparent for faces and small objects.

**Questions:**

The proposed method determines the starting point in the latent space to deviate the semantics of the objects. I think this search strategy is orthogonal to other editing methods that manipulate the latent based on the textural prompt. Hence, I wonder whether the proposed method is compatible with other editing methods, like null inversion. I would like to hear the authors' feedback regarding this.

---

> ### Author Response · Authors · 2023-11-18
> **Response to Reviewer nVsr**
>
> __Q1: It is suggested to report the image editing speed.__
>
> A1: Our image editing speed table has been incorporated into Major Response Q1.
>
> __Q2: Some writing parts can be improved for better clarity.__
>
> A2: We comprehensively revise the paper to enhance clarity in expression.
>
> __Q3: Since the re-inversion step is always set to 20% of the total steps, the non-edit regions will inevitably be affected.__
>
> A4: Thank you for bringing out this issue. Throughout our experiments, we observed that the impact of the number of re-inversion steps is trivial, thus we fix it as 20% of the total steps. However, better results should be achieved with a more careful hyperparameter tuning.
>
> __Q4: Is the proposed method compatible with other editing methods?__
>
> A4: Our search metric can be combined with other inversion-based image editing methods to enhance their performance. Taking Null-text Inversion (NTI) as an example, the search metric can improve NTI in multiple ways. We introduce the compatible method in Appendix A.8 (Figure 13) and supplemented it with experiments. In Appendix A.8, we supplement experiments using the search metric to find the hyperparameters for Null-text Inversion. Additionally, we provide detailed explanations of various compatible methods.

---

> ### Author Response · Authors · 2023-11-20
> **Happy to provide additional clarification**
>
> We sincerely thank you again for your great efforts in reviewing this paper. We have addressed your major concerns about speed and integration with other inversion-based editing methods. Please don’t hesitate to let us know if you have any further questions.

---

### Official Review · Reviewer_r1AF · 2023-10-31

**Soundness:** 2 fair
**Presentation:** 3 good
**Contribution:** 2 fair
**Rating:** 3
**Confidence:** 3

**Summary:**

This paper introduces advancements in text-driven image editing using diffusion-based methods. Existing techniques involve a fixed number of inversion steps to edit images aligned with a target prompt, but the optimal number of steps varies for different editing pairs. The paper introduces a new approach called Object-aware Inversion and Reassembly (OIR) to enable fine-grained, object-level editing. The object-level fine-grained editing is achieved by segmentation from SAM. It uses a search metric to determine the optimal inversion step for each editing pair and combines them for the final edited image, demonstrating superior performance, especially in multi-object editing scenarios.

**Strengths:**

1. The paper is well written, and the idea is clearly demonstrated through figures.

2. Although it is natural to do some grid search on the inversion steps when doing image editing, this paper introduces a systematic and principled way to do the search with a quantitative metric.

3. The idea of working on each object separately and reassemble is a interesting way to do multi-object editing.

4. Results are competitive compared to other inversion based editing techniques.

**Weaknesses:**

1. The method introduces significant computational overhead. For each input image, it has to run a search for optimal denoising steps for  each edit pair. The search process can be very time consuming. as it requires denoising from each inversion step, plus a metric calculation step. The overall amount of computation for a single image editing task can thus be very large.

2. It relies on SAM to do fine-grained segmentation of the input image to localize the object, and then do localized editing followed by resembling. However, it has some limitations. For example, the segmentation may not work well for small object. In addition, using SAM constrained the edit to object-level, while the applicability of more global change is questionable (e.g., change the style of an image from spring to winter, change the background to Mars, etc.). It may not be able to change the location of an object.

3. Related to above point, the competitive method, such as null-text inversion, does not involve segmentation and localization, therefore it is a bit unfair to compare with them, as in principle they can only benefit from the segmentation and localization. This is independent from the main innovation of this paper, which is searching for optimal inversion step.

4. Overall, it is a method that combines multiple existing component, with significant increase in compute. Therefore, the contribution and significance of the method is limited.

**Questions:**

N/A

---

> ### Author Response · Authors · 2023-11-18
> **Response to Reviewer r1AF**
>
> __Q1: The search process can be very time consuming.__
>
> A1: We supplement the time overhead experiments and summarize our acceleration methods in the Major Response Q1.
>
> __Q2.1: It relies on SAM to do fine-grained segmentation.__
>
> A2.1:
> - SAM is not mandatory. Our method is complementary to various mask generation schemes, such as attention map-based mask extraction methods that do not require introducing additional models. These methods were employed by notable works such as DiffEdit [a] and MasaCtrl [b].
>
>
> - In the revised paper, we provide a comparison of different mask generators in Appendix A.7 (Figure 12). From the figure, it can be observed that our method is highly robust to different mask generators.
>
> [a] MasaCtrl: Tuning-Free Mutual Self-Attention Control for Consistent Image Synthesis and Editing, ICCV 2023.
>
> [b] Diffedit: Diffusion-based semantic image editing with mask guidance, ICLR 2023.
>
> __Q2.2: The applicability of more global change is questionable (e.g., change the style of an image from spring to winter, change the background to Mars, etc.).__
>
> A2.2:
> - We are applicable to global change and we have extensively demonstrated these results in our initial submission. For example, in the main text, Column h of Figure 4, Figure 6, and Figure 16 with tasks (b, k), (b, m), (c, l), (c, m), (d, m), as well as Figure 17 with tasks (a, h), (c, f), (d, f), (d, h), (e, g), (e, h), (e, i), all involve background editing. In Figure 16, tasks (c, k), (f, j), and in Figure 17, tasks (c, g), (c, i) all include style editing.
> - Background can be considered as a generalized object for editing. In tasks that involve changing the style, alterations can be made on different objects separately, and then our reassembly strategy can be utilized to achieve global interaction.
>
> __Q2.3: It may not be able to change the location of an object.__
>
> A2.3:
> - Given an input image $I$ and a target prompt $P$, image editing's goal is to generate a new image $O$ that complies with $P$ and preserves the structure and semantic layout of $I$. The task of changing the location of an object is not under the current setting. More details about the editing setting are thoroughly explained in the first paragraph of Section 3 of our paper and have also been discussed in literature, such as Prompt-to-Prompt [c] and Plug-and-Play [d].
>
> [c] Prompt-to-Prompt Image Editing with Cross Attention Control, ICLR 2023.
>
> [d] Plug-and-Play Diffusion Features for Text-Driven Image-to-Image Translation, CVPR 2023.
>
> __Q3: Related to the above point, the competitive method, such as null-text inversion, does not involve segmentation and localization, therefore it is a bit unfair to compare with them, as in principle, they can only benefit from segmentation and localization. This is independent from the main innovation of this paper, which is searching for optimal inversion step.__
>
> A3:
> - Null-text Inversion (NTI) is a training-based approach that introduces learnable null-text embeddings to address DDIM Inversion's inability to accurately reconstruct the original image. It needs to be combined with other editing methods to achieve image editing, so whether NTI needs a mask depends on the editing method with which it is combined.
> - Our method is tightly coupled with mask generation and is completely training-free. We employ an additional mask generator to disentangle objects, enabling object-aware editing by separating individual entities.
> - We combine NTI with Prompt-to-Prompt (P2P) as a comparative method by using Grounded-SAM as the mask generator in P2P, which is the same as our method. The experimental results presented in Appendix A.10 (Figure 15) indicate that our training-free method outperforms the NTI+P2P counterpart which requires training. Moreover, we have also compared our method with other mask-based editing methods, and stable diffusion inpainting, in Section 4.1 of the initial submission.
>
> [e] Null-text Inversion for Editing Real Images using Guided Diffusion Models, CVPR 2023.
>
> __Q4: The contribution and significance of the method is limited.__
>
> A4: Please refer to Major Response Q2.

---

> ### Author Response · Authors · 2023-11-20
> **Happy to provide additional clarification**
>
> We sincerely thank you again for your great efforts in reviewing this paper. We have addressed your major concerns about speed, contribution, the relation with SAM, and additional comparison with other methods using SAM. Please don’t hesitate to let us know if you have any further questions.

---

### Official Review · Reviewer_pgMn · 2023-10-31

**Soundness:** 4 excellent
**Presentation:** 3 good
**Contribution:** 4 excellent
**Rating:** 8
**Confidence:** 5

**Summary:**

This paper discovers that various editing pairs exhibit differing levels of editing complexity. Additionally, it is observed that neglecting the varying difficulty levels of different editing pairs in multi-object editing tasks results in the problems of concept mismatch and poor editing. To address these issues, the paper introduces a novel training-free image editing approach called OIR. This method follows the approach of assembly first and then reassembly. 1) In the assembly strategy, the paper introduces a novel search metric. This metric automatically identifies the optimal inversion step for different editing pairs, enabling automatic control of editing difficulty. This approach allows different editing pairs to undergo separate denoising, preventing concept mismatch issues. Moreover, employing the search metric to find the optimal result represents a new paradigm in single-object editing. 2) In the reassembly strategy, the article suggests merging the editing regions and non-editing region during the reassembly step. This operation takes place in the denoise latent space. The reassembly process incorporates a re-inversion strategy, enhancing the image editing's edge smoothness, improving image editability, and enabling interaction across regions. 3) To assess OIR's capabilities, the authors collect two datasets, which are employed to evaluate both the single-object editing proficiency of the search metric and the multi-object editing capability of OIR. Numerous experimental results indicate that the search metric performs on par with existing state-of-the-art editing methods in single-object editing tasks. Moreover, in multi-object editing tasks, OIR demonstrates strong performance, outperforming the previous SOTA methods

**Strengths:**

1.	This paper identifies a fundamental challenge in multi-object image editing tasks. Previous methods typically treat an entire image as a whole entity during multi-object editing, without considering that editing pairs may have varying levels of editing complexity and therefore require different optimal inversion steps.
2.	The new search metric introduced in this article is simple yet effective. The approach of adding the two evaluation indicators makes sense, and visual verification confirms that the search metric aligns with the editing effect. In single-object editing tasks, employing the search metric yields promising results, comparable to other image editing methods.
3.	The OIR introduced in this article is novel and effective, presenting a new solution for the multi-object editing task. Unlike previous approaches, which treated the entire image as a whole, OIR breaks down the task into editing pairs. The experimental comparisons with other methods are extensive and thorough.
4.	The paper's structure is well-organized and easy to follow. The figures are well-designed, effectively illustrating the ideas and claims presented.

**Weaknesses:**

1.	In [a], it is mentioned that the inversion step can be considered as a fixed hyperparameter. However, the author only presents the results of the optimal inversion steps for example images, without demonstrating the overall distribution trend of the optimal inversion step across the entire dataset. Moreover, what are the distinct characteristics of editing pairs with larger and smaller optimal inversion steps respectively?
2.	In the reassembly strategy of OIR discussed in this paper, both reassembly step and re-inversion are mentioned. The paper indicates that both methods can smooth the edges of images and enable global information interaction. Re-inversion first inverts the spliced latent and then denoises it, essentially increasing the denoise step. Is it possible to replace the operation of re-inversion by increasing the reassembly step?
[a] SDEdit: Guided Image Synthesis and Editing with Stochastic Differential Equations, ICLR 2022.

**Questions:**

see weakness

---

> ### Author Response · Authors · 2023-11-18
> **Response to Reviewer pgMn**
>
> __Q1.1: What is the overall distribution trend of the optimal inversion step across the entire dataset?__
>
> A1.1: We collect 200 editing pairs from our multi-object dataset and employ the search metric to identify the optimal inversion step for each editing pair. The distribution of the optimal inversion steps is presented in Appendix A.9 (Figure 14). According to the figure, the majority of the optimal inversion steps occur within the range of 25 and 45.
>
> __Q1.2: What are the distinct characteristics of editing pairs with larger and smaller optimal inversion steps respectively?__
>
> A1.2: We notice that when modifying backgrounds or objects with substantial changes in shape, such as the sky or the ground, larger optimal inversion steps are required. On the other hand, situations involving objects and targets with similar shapes generally require smaller inversion steps. Appendix A.9 (Figure 14) displays several cases within these scenarios.
>
> __Q2: Is it possible to replace the operation of re-inversion by increasing the reassembly step?__
>
> A2:
> The re-inversion can not be replaced by increasing reassembly steps for the following reasons.
> - The reassembly steps cannot be increased arbitrarily. As detailed in Section 3, it must be smaller than the smallest optimal inversion step in all editing pairs, to preserve the original features of non-editing regions. This limits the ability to improve editing results by changing the reassembly step. By contrast, the number of re-inversion steps is free from such limitations.
> - Re-inversion improves interactions between various regions. Because re-inversion can extend the denoise steps, it enhances the global information exchange between various regions, leading to more realistic and natural outcomes.

---

> > ### Comment · Reviewer_pgMn · 2023-11-22
> >
> > Thanks authors for providing a nice response  of my review and most of my concerns are addressed.

---

> ### Author Response · Authors · 2023-11-20
> **Happy to provide additional clarification**
>
> We sincerely thank you again for your great efforts in reviewing this paper. We have addressed your major concerns about the distribution of optimal inversion step and the importance of re-inversion. Please don’t hesitate to let us know if you have any further questions.

---

### Official Review · Reviewer_xVir · 2023-11-01

**Soundness:** 3 good
**Presentation:** 3 good
**Contribution:** 2 fair
**Rating:** 5
**Confidence:** 2

**Summary:**

The paper proposes object-aware inversion and reassembly for image editing. The motivation is that the inversion steps vary from the editing of different objects. Therefore, we need to choose different inversion steps for different objects. Also, for different objects in one image, we need to merge the editing results. Then the reassembly strategy is introduced. The proposed method achieves state-of-the-art performance.

**Strengths:**

1) The paper is well-organized and easy to follow.

2) It makes sense to use different diffusion steps for different objects when performing the image editing.

3) The paper proposes a reassembly strategy to merge the editing results.

4) The proposed method achieves state-of-the-art performance.

**Weaknesses:**

1) To determine the optimal inversion steps for image editing, we need to inverse the image to all steps and then edit them accordingly. It is time-consuming and not automatic.

2) The two contributions are more like the engineering stuff. However, admittedly, they do bring a lot of performance gain.

**Questions:**

Please see my concerns in the weakness part.

---

> ### Author Response · Authors · 2023-11-18
> **Response to Reviewer xVir**
>
> __Q1: The method is time-consuming and not automatic.__
>
> A1: We supplement the time overhead experiments and summarize our acceleration methods in the Major Response Q1.
>
>
> __Q2: The two contributions are more like engineering stuff.__
>
> A2: We identify a significant challenge in multi-object image editing and highlight that different editing pairs require distinct optimal inversion steps. Our two contributions, including a new search metric and an object-aware inversion and reassembly process, are specifically designed to enable object-level fine-grained control, remarkably outperforming previous SOTA methods. More details can be found in Major Response Q2.

---

> > ### Comment · Reviewer_xVir · 2023-11-23
> > **Post Rebuttal Comments**
> >
> > Dear Authors,
> >
> > Thanks for your clarification. For the time cost for other methods, may I ask if they are implemented under one GPU or multiple GPUs?
> >
> > Thanks,

---

> ### Author Response · Authors · 2023-11-20
> **Happy to provide additional clarification**
>
> We sincerely thank you again for your great efforts in reviewing this paper. We have addressed your major concerns about contribution and speed. Please don’t hesitate to let us know if you have any further questions.

---

> ### Author Response · Authors · 2023-11-23
> **Response to Reviewer xVir**
>
> Dear Reviewer xVir,
>
> Other methods run on a single GPU because they lack clear parallel acceleration solutions due to the temporal dependency between the denoise steps. We use 2 GPUs for parallelization, in the table of Major Response Q1.
>
> Thanks.

---

### Author Response · Authors · 2023-11-18
**Major Response**

__Q1: The method is time-consuming (Reviewers xVir, r1AF and nVsr).__

A1:

| Method  | Ours (2 GPUs) | Ours (1 GPU) | Null-text Inversion  | Plug-and-Play | Stable Diffusion Inpainting | DiffEdit  |
| --- | --- | --- | --- | --- | --- | --- |
| Average time cost | 155s | 298s  | 173s  | 224s  | 6s  | 15s |
| Maximum GPU usage | 19.75 GB | 19.75 GB | 22.06 GB | 7.31 GB  | 10.76 GB | 13.51 GB |


- We have introduced a parallel acceleration solution for our method in Appendix A.4 of the initial submission, which is recognized by Reviewer nVsr.  In the new version of the paper, we move the details of the parallel acceleration method into "Acceleration for generating candidate images" in Section 3.1, and these details are highlighted in blue.
- We observe that the sequential optimal inversion step searching process in OIR is time-consuming. To address this issue, we propose a splicing strategy over the denoising process. Firstly, we pair different denoising processes with varying inversion steps to achieve equal total steps. For example, the denoising processes with 1 and 50 inversion steps are paired. Similarly, the 2 and 49-step inversions are paired.  In this way, denoising processes of the same total length can proceed simultaneously for parallel acceleration, as there is no dependency between denoising processes.  In addition, the optimal inversion step search of different editing pairs can be parallelized on multiple GPUs to achieve further acceleration.
- The table above presents a comparison results of editing speeds. It reveals that our method can be significantly accelerated in multi-GPU scenarios, whereas other methods lack clear parallel acceleration solutions due to the temporal dependency between the denoising steps. Detailed content is presented in blue font in Appendix A.6.


__Q2: The contribution and significance of the method is limited (Reviewer r1AF).__

A2:
We identified and addressed a fundamental challenge in multi-object image editing tasks, where existing methods typically apply the same inversion steps for all editing pairs, resulting in suboptimal editing quality. To this end, we propose a new editing paradigm to enable object-aware fine-grained editing, which integrates two tightly coupled components:

- A simple yet effective new search metric that automatically and dynamically selects the optimal inversion step for each editing pair, unlike previous inversion-based methods that set the inversion step as a fixed value. This approach effectively balances the trade-off between editability and fidelity. Moreover, we can easily accelerate this process by splicing denoising processes of varying lengths into equal lengths and running denoising processes of the same length in parallel.

- A novel disassembly and reassembly paradigm that is specially designed to disentangle the denoising process for each editing pair, which is not featured in prior works.
The significance and contributions of our paper have been well recognized by Reviewer pgMn and Reviewer nVsr.

---

### Author Response · Authors · 2023-11-18
**General response**

We sincerely thank all reviewers for their valuable comments.

## Novelty
Reviewer pgMn, Reviewer nVsr and Reviewer r1AF well recognize the novelty of our method:
- "*This paper identifies a fundamental challenge in multi-object image editing tasks. The new search metric introduced in this article is simple yet effective. The OIR introduced in this article is novel and effective, presenting a new solution for the multi-object editing task.*" (Reviewer pgMn)
- "*This paper identifies an interesting and useful phenomenon that the structure of different objects corresponds to a different level of latent as modeled in the diffusion process. Since the denoising process of different image parts is disentangled, the proposed method supports fine-grained multiple object editing, which is not featured in prior works.*" (Reviewer nVsr)
- "*The idea of working on each object separately and reassemble is a interesting way to do multi-object editing.*" (Reviewer r1AF)

## Promising Results
All reviewers agree that:
- "*The proposed method achieves state-of-the-art performance. They do bring a lot of performance gain.*" (Reviewer xVir)
- "*Employing the search metric yields promising results, comparable to other image editing methods. The experimental comparisons with other methods are extensive and thorough.*" (Reviewer pgMn)
- "*Results are competitive compared to other inversion based editing techniques.*" (Reviewer r1AF)
- "*The image editing results are indeed impressive, as shown in the main text and the appendix. Both the quantitative and qualitative results demonstrate the advantage over strong prior works.*" (Reviewer nVsr)

## Summary of changes
We have revised our submission and summarized our updates as follows:
1. We have provided image editing speeds for different methods in Appendix A.6. (Reviewer xVir, r1AF and nVsr)
2. We have presented the image editing results of our method under different mask generators in Appendix A.7. (Reviewer r1AF)
3. We have supplemented the comparison results between our method and the inversion-based method with a segment model in Appendix A.10. (Reviewer r1AF)
4. We have provided the experiment to visualize the distribution of optimal inversion steps in Appendix A.9. (Reviewer pgMn)
5. We have presented the combined approach and the experimental results of our method with the inversion-based image editing method in Appendix A.8. (Reviewer nVsr)
6. We have revised the paper to enhance clarity in expression. (Reviewer nVsr)

---

### Comment · Area_Chair_Cxa2 · 2023-11-20

Dear reviewers,

As the Author-Reviewer discussion period is going to end soon, please take a moment to review the response from the authors and discuss any further questions or concerns you may have.

Even if you have no concerns, it would be helpful if you could acknowledge that you have read the response and provide feedback on it.

Thanks,
AC

---

> ### Comment · Reviewer_pgMn · 2023-11-22
>
> Hi AC,
>     Since there are some contradictory  reviews  for this paper, especially for Reviewer r1AF, which vote 'no' to this paper. Do you want us to have any discussions on that ?

---

### Meta-Review · Area_Chair_Cxa2 · 2023-12-05

**Metareview:**

This paper proposes object-aware inversion and reassembly for image editing. The main idea is that different editing pairs (i.e., source and target) requires different inversion steps. The proposed method searches for the optimal inversion step for each subject pair, and reassemble the output. Reviewers think that the proposed method leads to performance gain, but introduces computational overhead. There are no consensus reached in the discussion period. After going through the paper, reviews, and responses, the AC thinks that the strengths outweighs the weaknesses. Therefore, an acceptance is recommended. The authors are advised to include the discussion of the computational overhead in the main paper to maximize the impacts brought to the community.

**Justification For Why Not Higher Score:**

Despite being effective, the weaknesses, including computational overhead and the reliance on on-the-fly models, is non-negligible. Therefore, the AC believes that a poster would be most suitable for this paper.

**Justification For Why Not Lower Score:**

After reading the material, the AC believes that this paper deserves an acceptance due to its promising performance. It could bring insights to the community.

---

### Decision · Program_Chairs · 2024-01-16

Accept (poster)